# A unified theory of feature learning in RNNs and DNNs

**Jan P. Bauer** [1]   **Kirsten Fischer** [2]   **Moritz Helias** [2]   **Agostina Palmigiano** [1]

## Abstract

Recurrent and deep neural networks (RNNs/DNNs) are cornerstone architectures in machine learning. Remarkably, RNNs differ from DNNs only by weight sharing, as can be shown through unrolling in time. How does this structural similarity fit with the distinct functional properties these networks exhibit? To address this question, we here develop a unified mean-field theory for RNNs and DNNs in terms of representational kernels, describing fully trained networks in the feature learning, or maximal update parametrization ($\mu$P) regime. This theory casts training as Bayesian inference over sequences and patterns, directly revealing the functional implications induced by the RNNs' weight sharing. In DNN-typical tasks, we identify a phase transition when the learning signal overcomes the noise due to randomness in the weights: below this threshold, RNNs and DNNs behave identically; above it, only RNNs develop correlated representations across timesteps. For sequential tasks, the RNNs' weight sharing furthermore induces an inductive bias that aids generalization by interpolating unsupervised time steps. Overall, our theory offers a way to connect architectural structure to functional biases.

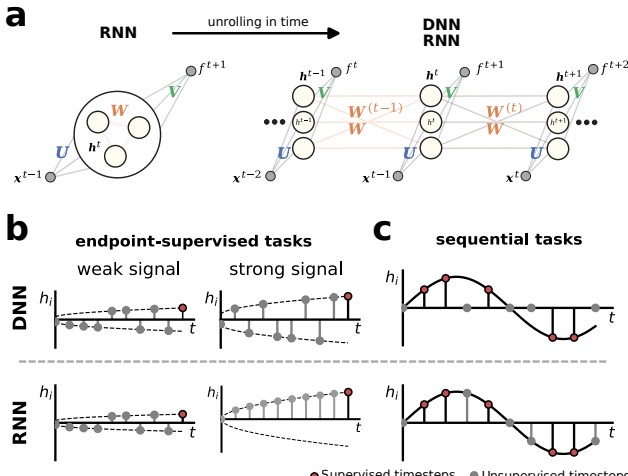

**Figure 1. Graphical abstract: RNNs resemble generalized DNNs after unrolling-in-time, but differ functionally depending on signal strength and tasks.** **a)** Recurrent neural network (RNN, left) and its unrolling-in-time representation (right): the same recurrent weights $W$ are shared across timesteps. **b)** *Endpoint-supervised tasks*: Supervision target at the last layer (red) affects hidden layer representation (gray), but only in the RNN and for sufficient signal strength induces a phase transition towards temporal coherence. **c)** *Sequential tasks*: The RNNs weight sharing induces a temporally coherent inductive bias that facilitates generalization from supervised (red) to unsupervised timepoints (gray), whereas the DNN exhibits "regression to the mean" for unsupervised points due to an effectively white prior.

## 1. Introduction

Understanding how neural networks learn useful representations from data remains a central question in machine learning theory. While recent work has studied feature learning in deep neural networks (DNNs) ([Yang & Hu, 2020](); [Bordelon & Pehlevan, 2022b](); [Cui et al., 2023](); [Seroussi et al., 2023](); [Fischer et al., 2024]()), the corresponding theory

for recurrent neural networks (RNNs) is far less developed. RNNs process sequences by repeatedly applying the same transformation over time, so that each new hidden state is computed using the same set of recurrent weights $W$. If we "unroll" this computation in time, the RNN can be viewed as a deep feedforward network whose layers correspond to timesteps. From this view, the two network architectures appear remarkably similar ([Fig. 1]()) and differ only in that the weights are shared across all layers (timesteps) in the RNN, whereas a standard DNN has layer-specific weights. It is so far unknown how weight sharing affects computation: are the extra, untied parameters of a DNN always advantageous, or can the inductive bias imposed by weight sharing actually improve learning by better aligning the model with the structure of a task? Answering this requires a framework that places RNNs and DNNs on a common footing, so that the

---

[1]Gatsby Computational Neuroscience Unit, UCL, London, UK [2]IAS-6, Forschungszentrum Juelich, Juelich, Germany. Correspondence to: Jan P. Bauer, Agostina Palmigiano <{jan.bauer, a.palmigiano}@ucl.ac.uk>.

*Proceedings of the $43^{rd}$ International Conference on Machine Learning*, Seoul, South Korea. PMLR 306, 2026. Copyright 2026 by the author(s).

functional consequences of weight sharing can be isolated.

Our key contributions towards this question are:

- We develop a unified mean-field theory for RNNs and DNNs in terms of representational kernels, describing trained networks after convergence of stochastic gradient Langevin dynamics (SGLD) in the feature learning ($\mu$P) regime. This theory casts network training as Bayesian inference over timesteps and patterns, revealing the functional implications of their architectural structure.

- For tasks that are typical for DNNs ("*endpoint-supervised*" tasks), we analytically show that even within $\mu$P scaling, different phases of feature learning exist. Below a critical signal strength, the representational kernels of RNNs and DNNs coincide, although they differ from the kernels obtained in a lazy learning phase (neural network Gaussian process, NNGP), which characterize initialization. Beyond a critical signal strength, however, the RNN's representation enters a *temporally coherent* phase that is absent in the DNN.

- For tasks that are typical for RNNs ("*sequential*" tasks), we show how the RNN's weight sharing induces an inductive bias that enables more sample-efficient learning. This implies a functional advantage beyond the representational changes of the endpoint-supervised case, and underlines that task-model alignment not only depends on expressivity, but also on how well the architecture matches the structure of the task.

The code to reproduce all figures is available at github.com/japhba/temporal-feature-learning.

## 2. Related work

**Probabilistic descriptions of neural networks.** The study of feedforward neural networks in the infinite-width limit has a rich history, beginning with the connection to Bayesian inference (Neal, 1996; Lee et al., 2018). These foundational works established that infinitely wide neural networks with random weights under the standard parametrization (SP, all weights scale as $1/\sqrt{\text{width}}$) converge to Gaussian processes. More recently, this line of work has been generalized using mean-field methods to account for feature learning either by adopting $\mu$P scaling (Yang & Hu (2020), readout weights scale as $1/\text{width}$), or by considering proportional limits of width and number of data samples. Such theories have been formulated in terms of adaptive weight scales Li & Sompolinsky; Ariosto et al.; Pacelli et al., task-aligned readout weights (van Meegen & Sompolinsky, 2025), or kernel matrices (Seroussi & Ringel, 2021; Zavatone-Veth & Pehlevan, 2021; Fischer et al., 2024;

Lauditi et al., 2025), providing a probabilistic description of representations that goes beyond dynamics near initialization as described by the neural tangent kernel (NTK) (Jacot et al., 2018). We here develop a generalization of such kernel theories to RNNs in a way that treats RNNs and DNNs on equal footing.

**Bayesian inference in state-space models.** Our approach to analyzing recurrent networks directly connects to Bayesian inference in classical probabilistic models. In particular, for linear activation, the RNN architecture we consider is identical to a linear Gaussian state-space model with vanishing noise in the forward pass (Kalman, 1960). Whereas these models consider inference at fixed weights via Kalman filtering, we here consider a posterior over weights due to training as well, leading to a more general theory.

**Theory of recurrent neural networks.** The dynamics of randomly-coupled recurrent neural networks have been studied extensively. Dynamical mean field theory has been derived in networks with random unstructured (Sompolinsky et al., 1988; Molgedey et al., 1992; Toyoizumi & Abbott, 2011), excitatory and inhibitory (Kadmon & Sompolinsky, 2015; Sanzeni et al., 2023; Mastrogiuseppe & Ostojic, 2017) and low-rank (Mastrogiuseppe & Ostojic, 2019; Landau & Sompolinsky, 2018) connectivity (Schuecker et al., 2018; Segadlo et al., 2022; Schuessler et al., 2024), including their response to perturbations (Sanzeni et al., 2023; Nguyen et al., 2025; Palmigiano et al., 2023) and computational properties (such as memory (Toyoizumi & Abbott, 2011; Schuecker et al., 2018; Pereira-Obilinovic et al., 2023)). The study of learning in RNNs is less thoroughly explored. The recurrent neural tangent kernel (RNTK) (Alemohammad et al., 2021) extends the NTK framework (Jacot et al., 2018) to recurrent architectures, providing insights into their training dynamics in the infinite-width lazy training regime. Proca et al. (2025); Bordelon & Pehlevan (2025) analyzed the learning *dynamics* in recurrent linear networks in the balanced regime. Similar to our work, Bordelon et al. (2024); Clark et al. (2026) consider the learning *equilibrium* in RNNs, but consider RNNs in continuous time with a leak term (Amari, 1972; Sompolinsky et al., 1988), identifying a transition to chaos in terms of the learning signal. In contrast, we here establish a discrete, joint theory of DNNs and RNNs and analytically study transitions in the models' representation and their inductive bias. An interesting step forward is to find out how these transitions relate.

**Generalization, architectural inductive biases and feature learning.** Kernel methods have been employed to study generalization (Canatar et al., 2021; Simon et al., 2023; van Meegen & Sompolinsky, 2025; Rubin et al., 2025), with tight links to the training of neural networks

and the role of inductive biases (Aiudi et al., 2025). Meanwhile, other work has explored the impact of initialization necessary for feature learning, in particular the role of small initialization (Saxe et al., 2013; Atanasov et al., 2021; Kunin et al., 2024; Tu et al., 2024). We here study feature learning in terms of structured updates to the kernel's eigenvectors, highlighting the effects of architecture and initialization.

**Learning dynamics versus convergence.** Previous work has studied the *dynamics* of learning (Saxe et al., 2014; Bordelon & Pehlevan, 2025; Proca et al., 2025), by approaches including *dynamical* mean-field theory (Bordelon & Pehlevan, 2022b), but comes at the cost of making simplifying assumptions (Saxe et al., 2014; Proca et al., 2025) or complicated expressions. We here join a line of recent work that instead seeks out to only describe the network state after training when the weights have converged to a stationary distribution, significantly simplifying the theory (Seung et al., 1992; Cohen et al., 2021; Seroussi et al., 2023; Cui et al., 2023; Fischer et al., 2024; van Meegen & Sompolinsky, 2025; Lauditi et al., 2025).

## 3. Results

### 3.1. Unified feature learning theory for RNNs and DNNs

#### 3.1.1. SETUP

We consider a general architecture with hidden dimension $N$ and input dimension $D$. At each timestep $t$, the pre-activations $\boldsymbol{h}^t \in \mathbb{R}^N$ receive external temporally-dependent input $\boldsymbol{x}^{t-1} \in \mathbb{R}^D$ through read-in weights $\boldsymbol{U} \in \mathbb{R}^{N \times D}$ and recurrent inputs $\phi(\boldsymbol{h}^{t-1})$ through hidden weights $\boldsymbol{W}^{(t)} \in \mathbb{R}^{N \times N}$ for DNNs and $\boldsymbol{W} \in \mathbb{R}^{N \times N}$ for RNNs, where $\phi(\circ)$ is an elementwise activation function. The scalar output of the architecture $f^{t+1}$ is obtained through readout weights $\boldsymbol{V} \in \mathbb{R}^{1 \times N}$, so that the whole architecture reads

$$\begin{aligned} \boldsymbol{h}^t &= \boldsymbol{W}^{(t-1)}\phi(\boldsymbol{h}^{t-1}) + \boldsymbol{U}\boldsymbol{x}^{t-1}, \qquad (1) \\ f^{t+1} &= \boldsymbol{V}\phi(\boldsymbol{h}^t), \end{aligned}$$

$$\boldsymbol{W}^{(t)} = \begin{cases} \boldsymbol{W}, & \text{RNN} \\ \boldsymbol{W}^{(t)}, & \text{DNN} \end{cases}$$

with $t = 1, \ldots, T-1$. We will abbreviate this penultimate timestep with $T_- := T - 1$ from here on.

We group all parameters into $\Theta$ and abbreviate $\boldsymbol{\phi}^t := \phi(\boldsymbol{h}^t)$. This definition encompasses RNNs, "generalized" DNNs with input and output at arbitrary timepoints, and also standard DNNs, where $\boldsymbol{x}^0$ is the input and $f^T$ the output (see Section A.3 for details).

We optimize this model via gradient descent on a mean-squared loss $\mathcal{L}(\Theta; y, \boldsymbol{x}) = \frac{1}{2}\frac{1}{P|\mathcal{T}|}\sum_{p=1}^P \sum_{t \in \mathcal{T}} (y_p^t - f_p^t(\Theta, \boldsymbol{x}))^2$ over samples $p = 1, ..., P$ and a number $|\mathcal{T}|$

of supervised timesteps $t \in \mathcal{T}$. For each gradient update, we allow for an i.i.d. Gaussian noise of strength $\sqrt{2\kappa}$ and an independent weight decay of strength $\frac{\kappa}{G_{\boldsymbol{\theta}}}$, so that the overall update reads

$$\boldsymbol{\theta}_{s+1} = \boldsymbol{\theta}_s - \nabla_{\boldsymbol{\theta}} P|\mathcal{T}|\mathcal{L}(\Theta_s)\,\mathrm{d}s - \frac{\kappa}{G_{\boldsymbol{\theta}}}\boldsymbol{\theta}_s\,\mathrm{d}s + \sqrt{2\kappa\,\mathrm{d}s}\,\boldsymbol{\xi}_s. \quad (2)$$

This update is commonly referred to as SGLD (stochastic gradient Langevin dynamics) (Naveh et al., 2020). Note that in contrast to SGD where stochasticity arises from fluctuations between mini-batches, SGLD assumes noise that is i.i.d. across components and SGLD steps. It is possible to consider the limit $\kappa \to 0$, corresponding to deterministic full batch gradient descent (see A.2.1 and A.8).

It can be shown that after convergence this algorithm samples from a stationary distribution $\boldsymbol{\theta} \sim P(\Theta|y, \boldsymbol{x}) \propto \exp\{-P|\mathcal{T}|\mathcal{L}(\Theta; y, \boldsymbol{x})/\kappa - \frac{1}{2}\sum_{\boldsymbol{\theta}}\|\boldsymbol{\theta}\|^2/G_{\boldsymbol{\theta}}\}$ (Gardiner, 1985; Seung et al., 1992; Kardar, 2007). This distribution can likewise be interpreted as a Bayesian posterior combining two factors: first, a base distribution that is a Gaussian i.i.d. weight prior $P(\Theta) \propto \exp\{-\frac{1}{2}\sum_{\boldsymbol{\theta}}\|\boldsymbol{\theta}\|^2/G_{\boldsymbol{\theta}}\} \propto \mathcal{N}(\Theta|0, G_{\boldsymbol{\theta}})$, and a likelihood $\propto \exp\{-P|\mathcal{T}|\mathcal{L}(\Theta; y, \boldsymbol{x})/\kappa\}$, which may alternatively be regarded as a Bayesian regularization due to noisy labels, $y_p^t = f_p^t + \sqrt{\kappa}\xi_p^t$.

#### 3.1.2. THEORETICAL RESULT

We here sketch the conceptual steps to obtain our analytical results, which are detailed in Section A.3. The starting point is the partition function of the stationary distribution above, whose logarithm is the cumulant-generating function of the Bayesian posterior over the parameters $\Theta$. Since the parameters enter the forward pass (1) only linearly (conditional on the preceding layer's post-activation), we can carry out the Gaussian integrals over them analytically. This trades the high-dimensional weights $\Theta$ for a distribution over neural activations $h$, inputs $\boldsymbol{x}$, and labels $y$. This procedure reveals a natural *kernel* order parameter $\Phi$ (the representational kernel, defined below), together with a dual kernel $\tilde{C}$, which both concentrate in the large-$N$ limit. Their values are fixed self-consistently by a saddle point of the resulting action. Conditioned on these order parameters, the pre-activations factorize and are distributed as:

$$\begin{aligned} P(h|y, \boldsymbol{x}) \propto \exp\Big\{ &-\tfrac{1}{2}\mathrm{tr}\big[\mathbb{Y}_{\mathcal{T}}(v\Phi_{\mathcal{T}}^- + \kappa)^{-1}\big] \\ &-\tfrac{1}{2}h^{\intercal}(w[\![\Phi^-]\!] + u\mathbb{X}^-)^{-1}h + \phi(h)^{\intercal}\tilde{C}\,\phi(h)\Big\}. \quad (3) \end{aligned}$$

In making this change of variables, this theory summarizes the effect of weight learning in the form of the *kernel*

$$\Phi_{pp'}^{tt'} := \frac{1}{N}\sum_i^N \phi(h_{i,p}^t)\phi(h_{i,p'}^{t'}) \qquad (4)$$

$$\overset{N\to\infty}{\asymp} \quad \langle\phi(h_p^t)\phi(h_{p'}^{t'})\rangle_{P(h|y,\boldsymbol{x})}. \qquad (5)$$

In these expressions, contractions $a^\mathsf{T}b := \sum_p^P \sum_t^T a_p^t b_p^t$ are taken over the joint pattern-time space, and $(\circ^-)^{tt'} := \circ^{t-1,t'-1}$ is a shorthand to indicate time-shift. Likewise, $\mathbb{X}_{pp'}^{tt'} := \frac{1}{D}\sum_i^D x_{p,i}^t x_{p',i}^{t'}$ and $\mathbb{Y}_{pp'}^{tt'} := y_p^t y_p^{t'}$ denote input and label kernels, and $\mathbb{H}_{pp'}^{tt'} := \frac{1}{N}\sum_i^N h_{p,i}^t h_{p',i}^{t'} \asymp \langle h_p^t h_{p'}^{t'}\rangle_{P(h|y,\boldsymbol{x})}$ is the kernel for the pre-activations, defined in analogy to the kernel $\Phi$ for the post-activations in (4). A subscript $\circ_{\mathcal{T}}$ denotes the restriction of a kernel to the supervised timesteps $\mathcal{T}$, and $\Phi$, $\mathbb{H}$, and $\tilde{C}$ are matrices in $\mathbb{R}^{PT_- \times PT_-}$, with $T_- := T-1$ as defined before. Our theory shows that *in terms of the representation $P(h|y,\boldsymbol{x})$, RNNs and DNNs only differ by a masking operation in time*, $[\![\Phi^-]\!] := \left\{\begin{smallmatrix}\Phi^-, & \text{RNN}\\ \mathrm{diag}(\Phi^-), & \text{DNN}\end{smallmatrix}\right.$, capturing the effect of the DNN's weight independence in form of the correlation of the $h$-fields across timesteps. Intuitively, this masking reflects the statistics of the weights: in the DNN, the weights are independent across layers, and independent weights induce activations that are uncorrelated across timesteps, so that only the diagonal of the kernel survives. In the RNN, by contrast, the weights are shared and hence perfectly correlated across time, which allows correlations between activations at different timesteps to build up. We discuss the effect of this difference in the following sections.

As we show in Section A.3 in the large $N$ limit, the empirical averages $\frac{1}{N}\sum_i^N$ concentrate to a well-defined value. The conjugate variable $\tilde{C}(\mathbb{H},\Phi) = \tilde{C}_y + \tilde{C}_h$ encodes the two constraints that the activity $h$ is subjected to: $\tilde{C}_y = \frac{1}{2}v(v\Phi^- + \kappa)^{-1}\mathbb{Y}(v\Phi^- + \kappa)^{-1}$ aligns the representation towards the targets $y^t$, whereas $\tilde{C}_h = \frac{1}{2}w(w[\![\Phi^-]\!] + u\mathbb{X}^-)^{-1}\left(\mathbb{H} - (w[\![\Phi^-]\!] + u\mathbb{X}^-)\right)(w[\![\Phi^-]\!] + u\mathbb{X}^-)^{-1}$ acts as a force on $\phi(h)$ towards the prior describing the forward pass $\boldsymbol{h}^t = \boldsymbol{W}^{(t-1)}\boldsymbol{\phi}^{t-1} + \boldsymbol{U}\boldsymbol{x}^{t-1}$ with untrained weights. The scalars $u, w, v$ are intensive parameters that reparametrize the scales of the weights $G_{\boldsymbol{\theta}}$ so that the kernels become intensive (i.e., $\mathcal{O}(1)$ with respect to $N$) quantities that concentrate, and we likewise reparametrized $\mathsf{K}$ by the intensive quantity $\kappa$ (see Section A.3). In the limit of large networks, the replacement of the empirical average over neuron indices (4) with an average that factorizes over neurons (5) becomes exact; that is, the empirical average is self-averaging due to concentration.

In deriving (3), we traded the posterior distribution $P(\Theta|y,\boldsymbol{x})$ over parameters of extensive size in $N$ for a distribution $P(h|y,\boldsymbol{x}) = P(h|\Phi,\tilde{C},\mathbb{Y},\mathbb{X})$ over a single variable $h_i \equiv h$ which is identically distributed over neurons, and only depends on the kernels $\Phi, \tilde{C}, \mathbb{Y}, \mathbb{X}$. This theory straightforwardly recovers the $\mu$P scaling (Yang & Hu, 2020) for the prior variances as $(G_{\boldsymbol{U}}, G_{\boldsymbol{W}}, G_{\boldsymbol{V}}) = (u/D, w/N, v/N^2)$ as well as $\mathsf{K} = \kappa/N$, from two requirements: that pre-activations $h$ scale

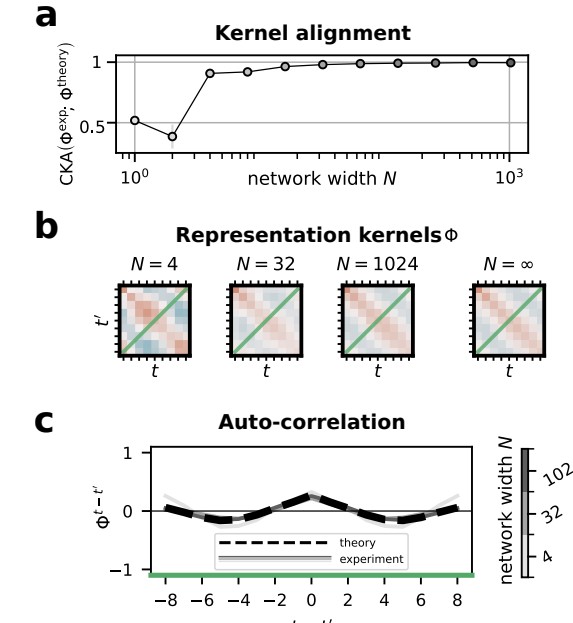

**a**

**Kernel alignment**

CKA($\Phi^{\exp.}, \Phi^{\text{theory}}$)

network width $N$

**b**

**Representation kernels $\Phi$**

$N = 4$   $N = 32$   $N = 1024$   $N = \infty$

$t'$ / $t$

**c**

**Auto-correlation**

$\Phi^{t-t'}$

- - - theory
—— experiment

$t - t'$

network width $N$

**Figure 2. Kernels in trained RNNs converge to theory predictions for large network width $N$.** We train an RNN as described in (2) with nonlinear activation function $\phi(\circ) = \mathrm{erf}(\frac{\sqrt{\pi}}{2}\circ)$ to produce a sinusoidal target sequence $y^t = \cos\left(\frac{2\pi}{T}t\right)$, $T = 10$ in response to a scalar input at time $t = 0$, i.e. $x^t = \delta^{t0}$. **a)** Centered kernel alignment (CKA) between the kernel $\Phi_{\exp.}(\Theta) = \frac{1}{N}\sum_i^N \phi(h_i)\phi(h_i)^\mathsf{T}$ from explicit weight SGLD experiments at different network widths $N$ and the kernel $\Phi_{\text{theory}} = \langle\phi(h)\phi(h)^\mathsf{T}\rangle_{P(h|y,\boldsymbol{x})}$ predicted by the theory (5) for $N \to \infty$. **b)** Temporal structure of kernels $\Phi_{\exp.}$ for different finite network widths compared to $\Phi_{\text{theory}}$ at infinite width $N \to \infty$. **c)** Autocorrelation function of the network measured along the anti-diagonal $\Phi^{t-t'}$ of the kernel (dashed: theory; full curve: numerics), marked in green in panel **b**.

suitably in the $N \to \infty$ limit (in particular avoiding exploding/vanishing gradients), and permitting feature learning in terms of a change to the representation kernel $\Phi$. Moreover, the theory will allow us to directly characterize representational and functional properties of the networks, which will be our focus in the following. We first test whether the theory can accurately capture the effects of weight sharing in a simple time-series regression task (see Fig. 2), before we leverage it in the following sections to explain the consequences of weight sharing. To this end, we compare kernels $\Phi_{\exp.}^{tt'}(\Theta) = \frac{1}{N}\sum_i^N \phi(h_i^t)\phi(h_i^{t'})$ measured from networks trained through SGLD experiments in weight space, and kernels $\Phi_{\text{theory}}^{tt'} = \langle\phi(h^t)\phi(h^{t'})\rangle_{P(h|y,\boldsymbol{x})}$ predicted by the theory (3). We use centered kernel alignment CKA($\Phi_{\exp.}(\Theta), \Phi_{\text{theory}}$) (Cortes et al., 2012; Fischer et al., 2024), which computes the cosine similarity of the vectorized matrices after subtracting their means. In Fig. 2a, the alignment between theory and simulations increases with the network width $N$ as expected, since

the empirical average we defined in (4) converges with the number of constituting terms. Notably, we obtain an accurate description of the network kernels already at large but finite network width.

The trained networks exhibit a non-trivial temporal correlation structure, visible as non-diagonal kernels in Fig. Fig. 2**b**; the anti-diagonal yields the autocorrelation function shown in Fig. 2**c**, which likewise converges to the theoretical prediction for large $N$. In this setting where there are no correlations in the input, there are potentially two contributions to the observed temporal coherence: the weight sharing over time and the temporal correlation in the supervision signal $y$. In the following sections, we use our theory to tease out these contributions, showing that such temporal correlations are specific to RNNs due to weight sharing across timesteps, which is absent in feed-forward architectures. In Section 3.2, we focus on *endpoint-supervised* tasks that are the typical use case of DNNs, and in Section 3.3 we focus on *sequential* tasks that are the typical use case of RNNs.

### 3.2. Phase transition towards temporal coherence in RNNs induced by strong learning signal in endpoint-supervised tasks

The formulation in (1) allows us to draw a direct comparison between DNNs and time-unrolled RNNs, with a one-to-one correspondence between RNN timesteps $t$ and DNN layers, which we thus also index by $t$. DNNs are typically trained on tasks with inputs provided only at the first layer and supervision applied only at the final layer. We refer to this setting as *endpoint-supervised*. The DNN equation (1) in this case becomes:

$$\boldsymbol{h}^1 = \boldsymbol{U}\boldsymbol{x}^0\,,\qquad\qquad (6)$$
$$\boldsymbol{h}^t = \boldsymbol{W}^{(t-1)}\phi(\boldsymbol{h}^{t-1})\,,\quad 2 \le t < T\,,$$
$$f^T = \boldsymbol{V}\phi(\boldsymbol{h}^{T-1})\,.$$

As we detail in Section A.3.4, DNNs can be understood as a special case of the general mean-field theory developed in Section 3.1 by omitting input at timesteps $t > 1$, and having no supervision target at timesteps $t < T$. Notably, DNNs in addition have independent weight priors $\boldsymbol{W}^{(t)}$ across timesteps, reflected in a factorization of the prior introduced in Section 3.1, $P_{\text{DNN}}(\{\boldsymbol{W}^{(t)}\}_t) = \prod_{t=1}^{T-1} P(\boldsymbol{W}^{(t)})$.

Despite these similarities, it is unclear what is the effect that weight sharing in the RNN has on representation (i.e., the kernels $\Phi$, $\mathbb{H}$) or on computation (i.e., the model output $f$). Segadlo et al. (2022) show that in the NNGP limit (i.e., when non-readout weights don't change during training), temporal correlations in RNNs with point-symmetric activation functions vanish, and the kernel representations in DNNs and RNNs therefore coincide. Here, we generalize their theoretical approach to account for feature learning under $\mu$P

scaling. Our theory reveals a distinction between the kernels in RNNs and DNNs in endpoint supervised tasks, but only if the learning signal is sufficiently strong (Fig. 3, here for simplicity for linear activation, i.e. $\phi(h) \to h$, $\Phi \to \mathbb{H}$; see Fig. A.7 for a simulation for the nonlinear case). We find a qualitative change in the RNN's kernel representation from a *temporally incoherent* to a *temporally coherent* regime that is absent in DNNs. We call a representation *temporally incoherent* when the kernel is diagonal in time, that is, when the pre-activation overlaps $\frac{1}{N}\boldsymbol{h}^t \cdot \boldsymbol{h}^{t'}$ between distinct timesteps $t \neq t'$ vanish (as is always the case in the DNN), and *temporally coherent* when these off-diagonal overlaps are non-zero, so that the pre-activation vectors at different timesteps align. We find that this transition is controlled by the strength of the learning signal $\lambda = \|y\|^2/vw^{T-2}u$ relative to the $\mathcal{O}(1)$ parameters $u, w, v$ that set the scale of the weights. This control parameter has the form of a signal-to-noise ratio: its numerator $\|y\|^2$ is the squared norm of the labels being learned, while its denominator collects the prior variances of the randomly initialized weights, which act as noise. We characterize this dependence more quantitatively below. We observe empirically that this transition is accompanied by an outlier eigenvalue in the spectrum of $\boldsymbol{W}$.

The pattern-by-pattern kernel $\mathbb{H}_{pp'}^{T_-T_-}$ in the last time point $T_- := T - 1$ determines the predictor statistics of the network via $f^T = v\mathbb{H}^{T_-T_-}(v\mathbb{H}^{T_-T_-} + \kappa)^{-1}y^T$ (Neal, 1996; Jacot et al., 2018). Interestingly, Fig. 3 reveals that the block structure induced by class membership is present in both RNNs and DNNs and differs only in scale, hence leading to qualitatively similar predictors. This reflects that both networks have learned the pattern-by-pattern structure $\mathbb{Y}$ that defines the task (Fig. 3**a**), in particular the last timestep's representation $\mathbb{H}_{pp'}^{T_-T_-} \simeq \mathbb{Y}_{pp'} = y_p y_{p'}^{\mathsf{T}}$ has learned $y$ as an eigenvector (cf. Fischer et al. (2024)).

We now seek a theoretical understanding behind this phenomenon. For simplicity, we here consider only the linear case (we show in Section A.8 that the transition also persists in the non-linear case). Starting from Eq. (3) that describes the probability distribution of the pre-activations, we obtain a probability over representations via $P(\mathbb{H}|y, \boldsymbol{x}) = \langle\delta(\mathbb{H} - hh^{\mathsf{T}})\rangle_{P(h|\Phi,\tilde{C},\mathbb{Y},\mathbb{X})}$ after replacing $\tilde{C} = \tilde{C}(\mathbb{H})$ by its stationarity condition (Section A.4):

$$-\ln P(\mathbb{H}|y, \boldsymbol{x})/N = \tfrac{1}{2}\text{tr}\left[\mathbb{Y}_{\mathcal{T}}(v\mathbb{H}_{\mathcal{T}} + \kappa)^{-1}\right]$$
$$+ \tfrac{1}{2}\text{tr}\left[\mathbb{H}(w[\![\mathbb{H}^-]\!] + u\mathbb{X}^-)^{-1}\right] - \tfrac{1}{2}\ln\frac{|\mathbb{H}|}{|w[\![\mathbb{H}^-]\!] + u\mathbb{X}^-|}\,,$$
$$(7)$$

where $[\![\mathbb{H}^-]\!] := \{\begin{smallmatrix}\mathbb{H}^-, & \text{RNN} \\ \text{diag}(\mathbb{H}^-), & \text{DNN}\end{smallmatrix}$ is, as before, the architecture-dependent masking and $|\circ|$ is the determinant. The second line takes the form of a Kullback-Leibler diver-

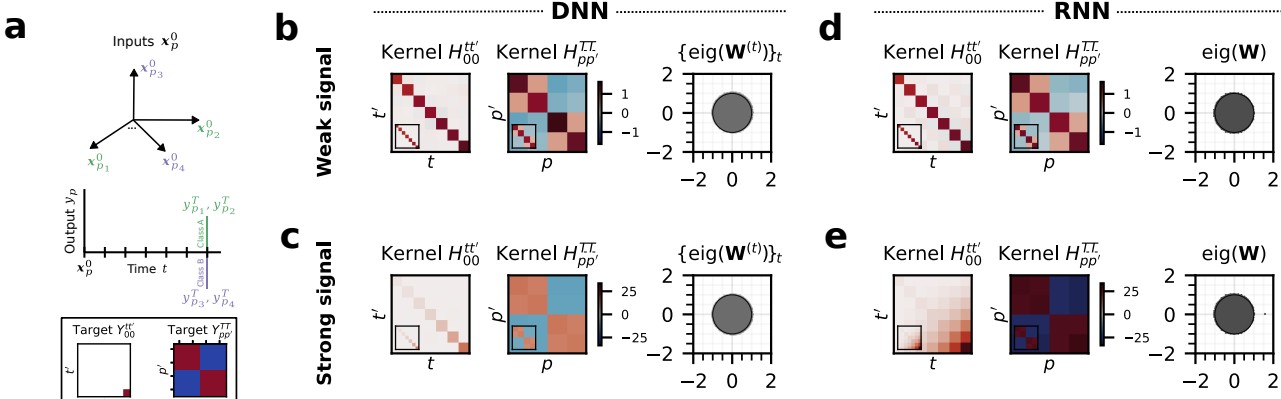

**Figure 3. RNNs and DNNs learn similar spatial representations while only RNNs learn temporal coherence for strong learning signal. a)** Binary classification task: $P = 4$ pairwise orthogonal inputs $\boldsymbol{x}_p \in \mathbb{R}^D$ with $D = 4$ map to labels $y_p \in \{-1, 1\}$, as summarized by kernels $\mathbb{Y}_{00}^{tt'} \in \mathbb{R}^{T_- \times T_-}$, $\mathbb{Y}_{pp'}^{TT} \in \mathbb{R}^{P \times P}$. **b-e)** Kernel and weight structure of DNNs (panels **b, c**) and RNNs (panels **d, e**) trained on the task, with $\phi(\boldsymbol{h}) = \boldsymbol{h}$. Lower-left insets show prediction by the kernel theory. We consider the cases of weak (upper row) or strong learning signal (lower row). From left to right in each cell (panels **b-e**): temporal kernel $\mathbb{H}_{00}^{tt'}$ for fixed pattern $p = 0$, sample kernel $\mathbb{H}_{pp'}^{T_-T_-} := \frac{1}{N}\boldsymbol{h}_p^{T_-} \cdot \boldsymbol{h}_{p'}^{T_-}$ in last timestep $T_- := T - 1 = 7$, and eigenspectrum in the complex plane of hidden weights $\boldsymbol{W}$ (RNN) or $\{\boldsymbol{W}^{(t)}\}_t$ (DNN, eigenspectra of all $\{\boldsymbol{W}^{(t)}\}_t$ plotted on same axis), with $N = 2048$. Other parameters: $w = v = u = 1$, $\kappa = 0.1$.

gence $D_{\mathrm{KL}}\big(\mathcal{N}_h(\mathbb{H}) \,\|\, \mathcal{N}_h(w[\![\mathbb{H}^-]\!] + u\mathbb{X}^-)\big)$ that vanishes at the NNGP prior which describes the forward propagation under random weights. For the endpoint-supervised tasks we consider here, the first term has support only on the last timestep, $(v\mathbb{H}_{\mathcal{T}}^- + \kappa)^{-1}\mathbb{Y}_{\mathcal{T}} \to (v\mathbb{H}^{T_-T_-} + \kappa)^{-1}\mathbb{Y}^{TT}$.

**Temporally-incoherent regime** For small learning signal $\lambda$, the representation kernels $\mathbb{H}$ of DNNs and RNNs coincide. The consistency of the diagonal solution with the saddle point equations for the RNN can be seen as follows: due to the masking operation in the DNN, a diagonal kernel is a stationary point. Inserting such a diagonal solution into the saddle point equations for the RNN, there are no terms that would cause off-diagonal correlations. The diagonal solution hence obeys the saddle point conditions of the RNN too. What is unclear, though, is whether this is the only stationary point. The RNN could have additional solutions with non-vanishing off-diagonal elements that attain a higher likelihood than the diagonal solution. Indeed for larger $\lambda$ this will be the case.

Since the output $f$ after learning depends only on the kernel $\mathbb{H}$, in the temporally-incoherent regime, both DNNs and RNNs yield the same predictions and thus have the same generalization properties.

**Temporally-coherent regime** An explicit analytical form for the kernel $\mathbb{H}$ can be derived for the minimal case of $T = 4$ (details in Section A.5). As we show there, the continuous change in the order parameter is the signature of a second-order phase transition with a critical exponent $\frac{1}{2}$, where the kernel's off-diagonal $\mathbb{H}^{T_-1, T_-2}$ forms the or-

der parameter controlled by $\lambda$. Fig. 4 shows this transition for $T = 4$ and for larger $T$. For the latter case, to the best of our knowledge, no explicit form is available. The transition prescribed by our mean-field theory is closely followed by the kernel measured from direct SGLD experiments on the weights $\boldsymbol{U}, \boldsymbol{W}, \boldsymbol{V}$ (solid curves and diamonds, respectively, in Fig. 4a). The phase transition thus casts learning as an optimization problem, but with the weight loss $\mathcal{L}(\Theta; y, \boldsymbol{x})$ replaced by a lower-dimensional objective over kernels, $-\ln P(\mathbb{H}|y, \boldsymbol{x})/N$, which is shown in Fig. 4. Depending on the control parameter $\lambda$, this kernel objective has a unique minimum only at the temporally-incoherent solution when $\lambda$ is small, but this minimum becomes unstable in favor of a new minimum at the coherent solution beyond the transition point (see Fig. 4b).

The analytics for $T = 4$ developed in Fig. 4 for (7) suggest an energy-entropy tradeoff underlying this transition. In both architectures, the label likelihood in the first line increases the final variance $\mathbb{H}^{T_-T_-}$ by tying it to $\mathbb{Y}^{TT}$ ("energy", cf. (11)). However, this incurs a penalty on $\mathbb{H}^-$ in $D_{\mathrm{KL}}\big(\mathcal{N}_h(\mathbb{H}) \,\|\, \mathcal{N}_h(w[\![\mathbb{H}^-]\!] + u\mathbb{X}^-)\big)$ that is the second line of (7), which encourages the kernels to follow the NNGP iteration over time. In the DNN case, the masking $[\![\mathbb{H}^-]\!] = \mathrm{diag}(\mathbb{H}^-)$ discards off-diagonals, so introducing temporal correlations cannot reduce the $D_{\mathrm{KL}}$. Thus, they remain suppressed by the $\ln|\mathbb{H}|$ term (an "entropic" penalty, since it favors uncorrelated $\mathbb{H}$). In the RNN however, the increase in $\mathbb{H}^{T_-T_-}$ can be compensated by $[\![\mathbb{H}^-]\!] = \mathbb{H}^-$ through the introduction of temporal correlations. As it turns out, the marginal cost of the entropic penalty from off-diagonals eventually drops below the $D_{\mathrm{KL}}$ cost, once the

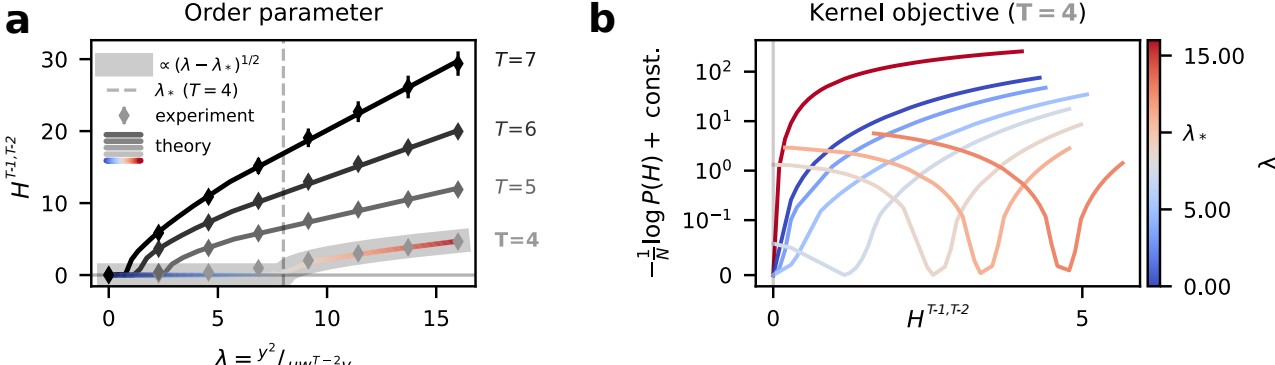

**Figure 4. Second-order phase transition in linear RNNs of $T = 4$ with critical exponent $\frac{1}{2}$. a)** Off-diagonal kernel order parameter $\mathbb{H}^{T\text{-}1,T\text{-}2}$ as a function of the control variable $\lambda$. Solid curves: kernel theory for different network depths. Diamonds: empirical kernel from weight SGLD. **b)** Negative log-probability for the off-diagonal order parameter $\mathbb{H}^-$, obtained from maximizing $P(\mathbb{H}|\mathbb{H}^{T\text{-}1,T\text{-}2}, y, \boldsymbol{x})$ as a function of the signal strength control variable $\lambda$. Network width $N = 2048$ (error bars: residual fluctuation at the equilibrium of the update (2)).

labels $\mathbb{Y}$ are sufficiently large relative to the priors $u, w, v$ that enter it. Thus, eventually a transition point controlled by their ratio $\lambda$ is reached, separating the phases.

### 3.3. RNN's weight sharing inductive bias generalizes efficiently for sequential tasks

In Section 3.2, we considered endpoint-supervised tasks, identifying qualitative difference in the representation across timesteps. While the last timestep's representation $\mathbb{H}^{T,T}$ exhibited quantitative differences between DNNs and RNNs in terms of its scale, its qualitative structure was preserved, entailing similar predictors $f$. Since such tasks do not read out from different timesteps and hence are unaffected by temporal correlations, this leaves the question how RNNs and DNNs differ for tasks with non-trivial temporal structure. To investigate this, we now consider sequential tasks $x^t \mapsto y^t$, where intuitively the RNNs' weight sharing may benefit generalization.

We consider a minimal setting where we have access to the ground truth learning signal $y^t = x_0^0 \sin(\omega t) + x_1^0 \cos(\omega t)$ for $\boldsymbol{x}^0 = [\cos(\varphi_0), \sin(\varphi_0)]^\mathsf{T} \in \mathbb{R}^{D=2}$, with a single input pattern ($P = 1$). Notably, this task is identical to a teacher RNN via $y^t = \boldsymbol{V}^\star (\boldsymbol{W}^\star)^t \boldsymbol{U}^\star \boldsymbol{x}^0$, with aligned readin $\boldsymbol{U}^\star = \left[\begin{smallmatrix} 1 & 0 \\ 0 & 1 \end{smallmatrix}\right]$, readout $\boldsymbol{V}^\star = \left[\begin{smallmatrix} 1 \\ 0 \end{smallmatrix}\right]$, and recurrent $\boldsymbol{W}^\star = \left[\begin{smallmatrix} \cos(\Delta\varphi) & -\sin(\Delta\varphi) \\ \sin(\Delta\varphi) & \cos(\Delta\varphi) \end{smallmatrix}\right]$ weights. The teacher RNN dynamics thus describe a rotation of the input vector $\boldsymbol{x}^0$ in a 2D plane by an angle $\Delta\varphi$ in each timestep. In Fig. 5, we find that RNNs trained on this task learn structured sequences from fewer samples (since $P = 1$, the number of supervised timesteps takes the role of a sample count). From a Bayesian perspective, the reason for this difference is the learned change in the kernel due to the masking operation $[\![\ldots]\!]$ in (3), which in turn determines the predictor via

$f^t = \sum_{t't''} \mathbb{H}^{tt'} ((\mathbb{H}+\kappa)^{-1})^{t't''} y^{t''}$, where summations now go over timesteps. In the DNN, the posterior covariance $\mathrm{diag}(\mathbb{H})$ will always produce an incoherent representation $h^t$ outside of the supervised training points, even after learning. This means that generalization effectively becomes Bayesian inference with an uncorrelated kernel, and thus extrapolating with the prior mean, $f^t = 0$ (in Fig. A.8 we show that this can have a regularizing effect if the task is *unstructured*). In contrast, the RNN's representation can develop correlations in $\boldsymbol{h}^t$ that match the target kernel $\mathbb{Y}$ already after a few samples ("task-model alignment"). In this regime, the trained recurrent matrix ($\boldsymbol{W}$) recovers the conjugate pair of teacher eigenvalues, indicating a a structured change in the spectrum. This difference in generalization is perhaps surprising, since due to its weight sharing, the RNN's expressivity is a subset of that of a DNN.

While we have shown that temporal correlations in $\mathbb{H}$ can develop, it remains unclear what is the mechanism that shapes them such that they indeed facilitate generalization. To understand this, we consider the kernel theory $P(\mathbb{H}|\mathbb{Y}, \mathbb{X})$ in (7) for the linear case and pursue a perturbative expansion in the label strength $\mathbb{Y}$. Starting from the closed-form saddle-point equation for the learned kernels (see (8), (34)), we expand around the diagonal NNGP kernel $\mathbb{H}_0$ (the solution in absence of feature learning) by writing $\mathbb{H} = \mathbb{H}_0 + \Delta$, with $\Delta = \Delta_1 + \Delta_2 + \mathcal{O}(\mathbb{Y}^3)$. The linear response $\Delta_1$ turns out to not involve strong architectural effects, since masking $[\![\mathbb{H}_0]\!]$ is an identity operation on diagonal kernels. We consider then the second order $\Delta_2$ to identify architecture-dependent effects. Introducing the NNGP observation covariance $\mathbb{G}_y := (v\mathbb{H}_0 + \kappa)^{-1}$, the leading correction can be

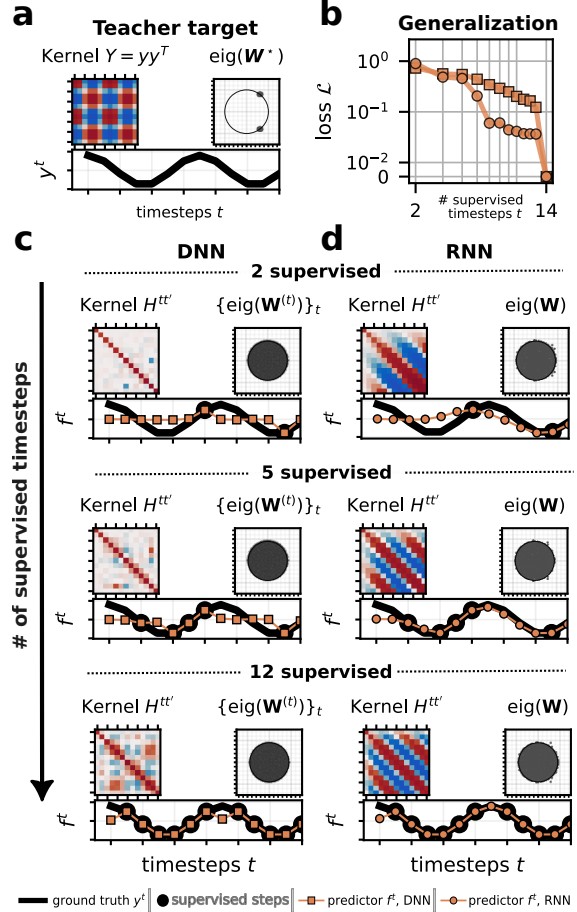

**a** Teacher target

Kernel $Y = yy^\mathsf{T}$  eig($W^\star$)

$y^t$

timesteps $t$

**b** Generalization

loss $\mathcal{L}$

$10^0$
$10^{-1}$
$10^{-2}$
$0$

2   # supervised   14
   timesteps $t$

**c** DNN

# of supervised timesteps

**2 supervised**

Kernel $H^{tt'}$   $\{\text{eig}(W^{(t)})\}_t$

$f^t$

**5 supervised**

Kernel $H^{tt'}$   $\{\text{eig}(W^{(t)})\}_t$

$f^t$

**12 supervised**

Kernel $H^{tt'}$   $\{\text{eig}(W^{(t)})\}_t$

$f^t$

timesteps $t$

**d** RNN

**2 supervised**

Kernel $H^{tt'}$   eig($W$)

$f^t$

**5 supervised**

Kernel $H^{tt'}$   eig($W$)

$f^t$

**12 supervised**

Kernel $H^{tt'}$   eig($W$)

$f^t$

timesteps $t$

— ground truth $y^t$ | ● supervised steps | —■— predictor $f^t$, DNN | —●— predictor $f^t$, RNN

**Figure 5. RNNs have better sample efficiency in sequential tasks due to task-model alignment induced by weight sharing.** **a)** Sequence regression task (label kernel $\mathbb{Y}$, teacher spectrum $W^\star$, and sinusoidal teacher output). **b)** Generalization error $\mathcal{L} = \frac{1}{2T} \sum_{t=2}^{T} (y^t - f^t)^2$ across all timesteps versus the number of supervised timesteps. **c-d)** Representation $\mathbb{H}$ and output $f^t$ for the DNN (**c**) and RNN (**d**) at different numbers of supervision steps (rows), with $\phi(h) = h$; each cell shows the time-by-time kernel, the weight eigenspectra, and the target $y^t$ versus prediction $f^t$.

written as

$$\mathbb{H} - \mathbb{H}_0 \approx +\mathbb{G}_y \mathbb{Y}^+ \mathbb{G}_y \qquad \Big\} = \triangle_1 \quad (8)$$

$$- \left( w^{-2}\, \mathbb{H}_0^{-1} [\![\triangle_1]\!]\, \mathbb{H}_0^{-1} \triangle_1 \mathbb{H}_0^{-1} + (\sim)^\mathsf{T} \right) + ... \quad \Big\} = \triangle_2$$

where $[\![...]\!]$ again is the identity for RNNs and a diagonal projection for DNNs, $\mathbb{Y}^+$ denotes time shift, $(\sim)^\mathsf{T}$ is the transpose of preceding term, and we omitted terms independent of masking "..." or that are of higher order $\mathcal{O}(\mathbb{Y}^3)$.

To interpret this expression, consider a time-diagonal $\mathbb{X}$ and a kernel $\mathbb{H}_0$ (Segadlo et al., 2022) with *constant* diagonal, conditions which are both met for Fig. 5 for $u = w = v = 1$. Then, $\mathbb{H}_0^+$ is diagonal with entries $\mathbb{h}_0^t$ and we may write $\triangle_1 = \mathbb{G}_y \mathbb{Y}^+ \mathbb{G}_y \propto \mathbb{Y}^+$.
In a DNN, $[\![\triangle_1]\!] = \text{diag}(\triangle_1)$ and the quadratic term acts

entry-wise,

$$\triangle_{2,\text{DNN}}^{tt'} = -\frac{1}{\mathbb{h}_0^t} \left( \frac{1}{\mathbb{h}_0^t} \mathbb{Y}^{+tt} + \frac{1}{\mathbb{h}_0^{t'}} \mathbb{Y}^{+t't'} \right) \mathbb{Y}^{+tt'} \frac{1}{\mathbb{h}_0^{t'}} + ..., \quad (9)$$

In an RNN, in contrast, we get

$$\triangle_{2,\text{RNN}}^{tt'} = -2 \frac{1}{\mathbb{h}_0^{t'}} \left( \sum_{t''} \mathbb{Y}^{+tt''} \frac{1}{\mathbb{h}_0^{t''}} \mathbb{Y}^{+t''t'} \right) \frac{1}{\mathbb{h}_0^{t'}} + ... . \quad (10)$$

This equation shows that the inverses appearing in (8) take the role of *propagators*, and hence (10) implements an interpolation between supervised points of $\mathbb{Y}$. To see this explicitly, consider the case that $t_1$ and $t_2$ have been observed but $t_3$ has not. Then, the correlation $\triangle_{2,\text{RNN}}^{t_3 t_1}$ will interpolate across the path $t_3 \leftarrow t_2 \leftarrow t_1$ that appears in the sum at $t' = t_2$, but $\triangle_{2,\text{DNN}}^{t_3 t_1} = 0$ in (9), making the posterior over $h$ uncorrelated.

A complementary way to understand this is that $\triangle_1$ is an $\mathbb{H}_0$-whitened version of $\mathbb{Y}^+$, and due to $\mathbb{H}_0$'s diagonal structure will approximately inherit its eigenvectors. For the DNN, the masking in the second-order iteration (9) will destroy this structure, so that learning requires more samples.

In summary, this shows that the mechanism behind the inductive bias of weight sharing can be traced back to the propagation of label messages across unsupervised time points.

## 4. Discussion

**Summary.** In this work, we developed a unified theory of feature learning for RNNs and DNNs in terms of representational similarity kernels $\Phi = \frac{1}{N} \sum_i^N \phi(h_i)\phi(h_i)^\mathsf{T}$ and $\mathbb{H} = \frac{1}{N} \sum_i^N h_i h_i^\mathsf{T}$, describing trained networks in the feature learning ($\mu$P) regime. This theory reveals the functional implications of architecture: it describes how the representational kernels $\Phi$ and $\mathbb{H}$ as well as the predictor $f$ are shaped by weight scales $u, w, v$, the structure of input $\mathbb{X}$, the labels $\mathbb{Y}$, and the network architecture itself. In this theory, the key architectural difference between RNNs and DNNs, weight sharing across timesteps (layers), becomes a masking operation $[\![\Phi]\!]$ on the temporal correlations of hidden activities $\phi(h)$ (see (3)) which ultimately affects generalization. This establishes a link from structure (the weights $W$) to function (the representation $h$ and predictor $f$).

**Phases of feature learning.** Feature learning is often characterized in terms of $\mathcal{O}(1)$ changes in the representation (Yang & Hu, 2020; Bordelon & Pehlevan, 2022a). We find that, within this regime, a sufficiently strong learning signal $\lambda$ is necessary in addition to drive *structured* alignment in RNNs: not merely changes in eigenvalues of the representational kernel, but alignment of its eigenvectors to the task covariance. This suggests a more granular view of feature

learning beyond uniform changes in scale (Canatar et al., 2021; Li & Sompolinsky, 2021): the learned eigenvectors themselves constitute the features that improve generalization, consistent with how the term "feature" is used in neuroscience (Hubel & Wiesel, 1962; Yamins et al., 2013). The phase transition we identify is reminiscent of a Baik-Ben Arous-Péché (BBP) transition (Baik et al., 2004), where random initialization acts as noise that gradient signals must overcome, mirroring similar findings by Bordelon & Pehlevan (2025).

**Connection to initialization.** In practice, feature learning is often controlled via initialization scale, small initialization leading to stronger representation of data features in the network learned. In our theory, the parameters $u, w, v$ control a Gaussian prior over weights, and a narrow prior (large $\lambda = \|y\|^2 / uw^{T-2}v$) drives the representation from *temporally incoherent* to *temporally coherent*. These two perspectives match in the small noise regime $\kappa \ll 1$, where the diffusion over the Gaussian prior becomes much slower than changes due to the loss. Our theory thus details why small initialization aids feature learning (Saxe et al., 2013; Atanasov et al., 2021; Tu et al., 2024), here extended to the temporal domain.

**Inductive bias and task-model alignment.** A central finding is that weight sharing endows RNNs with an inductive bias that can outweigh the DNN's greater expressivity. In sequential tasks, RNNs generalize from fewer samples due to a representation which infers the task structure, whereas DNNs effectively revert to a white prior outside supervised timesteps. This is in line with "No Free Lunch" theorems (Wolpert, 1996): the larger parameter count of DNNs need not be universally better when the inductive bias of a more constrained model matches the task structure. Our theory provides a mechanism behind this phenomenon in terms of interpolation between labels.

**Beyond recurrent networks.** Transformers lack the explicit weight sharing across layers that defines RNNs, yet they share their query, key and value projections across timesteps and can be mapped onto RNNs (Katharopoulos et al., 2020). A similar theoretical approach could thus be pursued for transformers: the property we expect to carry through is the factorization of pre-activations across neurons in the wide-network limit, which underlies our kernel description and is likewise expected from kernel limits of attention (Hron et al., 2020). Within such a description, positional encoding would induce correlations between timesteps analogous to the input correlations that enter our theory through the input kernel, rather than to the weight-sharing effects studied here. Whether gradient descent favors temporally coherent representations as in RNNs is an interesting open question.

**Limitations and future work.** Regarding scope, our theory deliberately chooses a functional perspective by marginaliz-

ing over the weights, thereby abstracting away their learning dynamics and structure. Regarding training, we approximated the correlated mini-batch fluctuations in plain SGD with i.i.d. noise of SGLD (Welling & Teh, 2011). Whereas the qualitative results discussed here persist in the noiseless case (see Section A.8), this replacement in general changes optimization trajectories. As noted, the limit $\kappa \to 0$ can be taken such that the prior variance $G_\Theta$ stays constant and the theory is still well defined, so that it describes full batch gradient flow. We have here focussed on the case of vanilla RNNs. It would be interesting to see how our findings carry over to architectures with a shared core, such as state-space models (Section A.8). A related challenge will be to incorporate gating, as employed by gated RNNs, into the present framework, possibly building on Krishnamurthy et al. (2022). Regarding the tasks considered, we considered minimal settings to identify foundational mechanisms of learning via interpretable analytics, notably linear networks and orthogonal inputs. This choice has allowed us to focus on temporal generalization. However, there are important other phenomena in deep learning that depend on the structure over patterns $p$ (Sclocchi et al., 2024) or their interaction with timesteps $t$, particularly in proportional limits like $T, P \propto N$. Likewise, representations will be shaped by the cumulative application of nonlinearities across layers (Keup & Helias, 2022). While in principle our theory describes these cases as well, they go beyond the scope of this work.

## Acknowledgements

We would like to thank Alexander van Meegen and Antonio Sclocchi for helpful discussions. This work is supported by the Gatsby Charitable Foundation (GAT3850 to JPB and AP), and the Simons Foundation (1156607 to AP) and by the Deutsche Forschungsgemeinschaft (DFG, German Research Foundation) - 368482240/GRK2416, the Helmholtz Association Initiative and Networking Fund under project number SO-092 (Advanced Computing Architectures, ACA), the Deutsche Forschungsgemeinschaft (DFG, German Research Foundation) as part of the SPP 2205 – 533396241, and the DFG grant 561027837/HE 9032/4-1 (to MH). The authors gratefully acknowledge the computing time granted by the JARA Vergabegremium and provided on the JARA Partition part of the supercomputer JURECA at Forschungszentrum Jülich (computation grant JINB33).

## Impact Statement

This paper presents work whose goal is to advance the field of Machine Learning. There are many potential societal consequences of our work, none which we feel must be specifically highlighted here.

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

# A. Appendix

## Overview

We structure the Appendix as follows: we first in Section A.1 establish the connection between Bayesian inference and the stationary distribution of stochastic gradient Langevin dynamics (SGLD). In Section A.3.1, we derive the infinite-width network prior and the resulting saddle-point equations for the network's kernel. We detail the effects of partial supervision in Section A.3.3 and the reduction to the standard DNN architecture in Section A.3.4. The section Section A.4 presents an analytical study of linear recurrent networks, deriving closed-form kernel solutions and performing a Landau analysis of the symmetry-breaking transition. Finally, we outline the numerical solution methods and provide a step-by-step summary of the derivation in Section A.7.

## A.1. Setup

Recall the definition of the RNN: at each time step $t$, the pre-activations are $\boldsymbol{h}^t \in \mathbb{R}^{N \times P}$, where $P$ is the index over training samples. The networks have one scalar output for each training sample $y^t \in \mathbb{R}^P$ and are updated according to

$$\boldsymbol{h}^t = \boldsymbol{W}\boldsymbol{\phi}^{t-1} + \boldsymbol{U}\boldsymbol{x}^{t-1} \quad 1 \leq t < T\,, \tag{11}$$
$$f^{t+1} = \boldsymbol{V}\boldsymbol{\phi}^t\,,$$
$$y^{t+1} = f^{t+1} + \xi^{t+1}\,,$$

where $\boldsymbol{x}^t$ is the input at time $t$ and we have defined the shorthand $\boldsymbol{\phi}^t = \phi(\boldsymbol{h}^t)$. The parameters $\boldsymbol{U} \in \mathbb{R}^{N \times D}$, $\boldsymbol{W} \in \mathbb{R}^{N \times N}$, $\boldsymbol{V} \in \mathbb{R}^{1 \times N}$ are matrices which we assume to have Gaussian priors that are i.i.d. over the matrices' entries

$$\boldsymbol{U} \overset{\text{i.i.d.}}{\sim} \mathcal{N}_{\boldsymbol{U}}(\boldsymbol{U}) := \mathcal{N}(0,\, G_{\boldsymbol{U}})\,, \tag{12}$$
$$\boldsymbol{W} \overset{\text{i.i.d.}}{\sim} \mathcal{N}_{\boldsymbol{W}}(\boldsymbol{W}) := \mathcal{N}(0,\, G_{\boldsymbol{W}})\,,$$
$$\boldsymbol{V} \overset{\text{i.i.d.}}{\sim} \mathcal{N}_{\boldsymbol{V}}(\boldsymbol{V}) := \mathcal{N}(0,\, G_{\boldsymbol{V}})\,,$$

for which our derivation will reveal the natural scalings $G_{\boldsymbol{U}} =: U = u/D$, $G_{\boldsymbol{W}} =: W = w/N$, $G_{\boldsymbol{V}} =: V = v/N^2$ with $\mathcal{O}(1)$ parameters $v, w, u$. The network output $f^t$ is a vector $\in \mathbb{R}^P$. Furthermore, we assume a readout noise that is i.i.d. in time and patterns, $\xi_p^t \overset{\text{i.i.d.}}{\sim} \mathcal{N}_{\mathsf{K}}(\xi_p^t) := \mathcal{N}(0, \mathsf{K})$ with $\mathsf{K} = \kappa/N$. We define time-advanced $(\circ^+)^{tt'} := \circ^{t+1,t'+1}$ and time-retarded $(\circ^-)^{tt'} := \circ^{t-1,t'-1}$ kernel matrices to ease notation. We also define the input kernel $\mathbb{X}^{tt'} := \frac{1}{D}\sum_{i=1}^D x_i^t x_i^{t'}$ and the label kernel $\mathbb{Y}^{tt'} := y^t y^{t'}$ for convenience.

## A.2. Bayesian inference and relation to training dynamics

We here derive a theory of learning that has two interpretations. The first is that of Bayesian inference. Second, as the stationary distribution of weights after training by stochastic gradient Langevin dynamics (SGLD). Both approaches will be described by the same set of equations.

**Bayesian inference** Given a set of training data $\mathcal{D} = \{(\boldsymbol{x}_p^{t-1}, y_p^{t+1})\}_{1 \leq p \leq P}^{1 \leq t < T}$, the Bayesian approach assumes a neuronal architecture, for example (11), which defines the network output as a function $f(\Theta, \boldsymbol{x})$ that depends on the network's input $\boldsymbol{x} \in \mathbb{R}^{T \times P \times D}$ and its parameters $\Theta$; in our case, $\Theta = \{\boldsymbol{W}, \boldsymbol{V}, \boldsymbol{U}\}$. The set of all inputs is combined into the matrix $\boldsymbol{x}$ and the set of outputs in a vector $y \in \mathbb{R}^{T \times P}$. One assumes a prior distribution $P(\Theta)$ on the set of parameters; in our case, the distribution $P(\Theta)$ is given by the set of distributions $\mathcal{N}_{\boldsymbol{U}}(\boldsymbol{U}), \mathcal{N}_{\boldsymbol{W}}(\boldsymbol{W}), \mathcal{N}_{\boldsymbol{V}}(\boldsymbol{V})$ (12).

Together with the observation likelihood $\mathcal{N}(y|f(\Theta, \boldsymbol{x}), \mathsf{K}\mathbb{I})$, this defines a posterior over weights

$$P(\Theta|y, \boldsymbol{x}) \propto \mathcal{N}(y|f(\Theta, \boldsymbol{x}), \mathsf{K}\mathbb{I})\, P(\Theta), \tag{13}$$

using the rules of Bayesian inference. In turn, this weight posterior induces a marginal distribution on labels $y$

$$P(y|\boldsymbol{x}) = \int P(y, \Theta|\boldsymbol{x})\, d\Theta. \tag{14}$$

We can obtain the statistics of any function $O(\Theta)$ under the posterior over weights from the cumulant generating function (sometimes referred to as "free energy")

$$\mathcal{W}(y,j) = \ln \int e^{j\,O(\Theta)}\,\mathcal{N}\big(y|f(\Theta,\boldsymbol{x}), \mathsf{K}\,\mathbb{I}\big)\,P(\Theta)\,d\Theta\,. \tag{15}$$

For example, the mean of $O$ follows as

$$\langle O(\Theta)\rangle_{P(\Theta|\boldsymbol{x},y)} = \partial_j\,\mathcal{W}(y,j)\big|_{j=0}\,,$$

where one notes that the outer derivative of the logarithm yields the correct normalization as in (13). Higher order cumulants are obtained as higher order derivatives.

**Stochastic gradient Langevin dynamics**  The second interpretation of (13) is that of a stationary distribution of a time-dependent learning rule for the weights. It is known that the stochastic differential equation

$$d\Theta(s) = -\nabla_\Theta H(\Theta)\,ds + dB(s)\,, \tag{16}$$

$$\langle dB(s)dB(s')\rangle = \delta(s-s')\,\frac{2}{\beta}\,ds\,,$$

where $dB(t)$ is a Wiener increment, has the stationary distribution (e.g., Gardiner (1985); Risken (1996))

$$P(\Theta) \propto \exp\big(-\beta\,H(\Theta)\big)\,. \tag{17}$$

For Gaussian prior measures on the weights with variance $g$ one has $P(\Theta) \propto \exp\big(-\|\Theta\|^2/2G\big)$, so that (14) can be written as

$$P(y|\boldsymbol{x}) \propto \int \exp\big(-\frac{1}{2\mathsf{K}}\|y - f(\Theta,\boldsymbol{x})\|^2 - \frac{1}{2G}\|\Theta\|^2\big)\,d\Theta\,,$$

so that we identify $H$ in (17) as

$$H(\Theta) = \frac{PT_-}{\mathsf{K}\beta}\,\mathcal{L}(\Theta) + \frac{1}{2G\beta}\|\Theta\|^2\,,$$

where the first term is expressed in terms of the the mean squared error between the given data and network output

$$\mathcal{L}(\Theta) = \frac{1}{2PT_-}\|y - f(\Theta,\boldsymbol{x})\|^2\,,$$

where the appearing vectors in the norm are $\mathbb{R}^{T\cdot P}$

Discretizing (16) in time and omitting a Metropolis-Hastings correction (MALA) Besag (1994) , this scheme becomes identical to plain stochastic gradient descent (SGD), with the difference that the source of the noise here is i.i.d. Gaussian, whereas it is more structured mini-batch noise in typical deployments of SGD.

### A.2.1. ORDERING OF THE NOISELESS AND STATIONARY LIMITS

A subtlety arises when taking the limit of Langevin noise to zero, which concerns the stationary distribution, because the limits $\lim_{\kappa\to 0}$ and $\lim_{s\to\infty}$ do not necessarily commute: the stationary state reached with gradient flow corresponds to $\lim_{s\to\infty}\lim_{\kappa\to 0}$, while considering the (stationary) Bayesian posterior distribution in the limit $\kappa \to 0$ corresponds to $\lim_{\kappa\to\infty}\lim_{s\to\infty}$. The two limits may not commute in cases in which ergodicity is broken in gradient flow; that is, if the training dynamics gets stuck within a local minimum. In Fig. A.11 and A.12 we empirically find that our results persist in this limit, presumably due to the absence of strict local minima/presence of directions of negative curvature in the loss landscape.

## A.3. Bayesian adaptive kernel theory for RNN and DNN

In this section we derive a unified mean-field formulation in kernel-space that treats deep feedforward (DNN) and recurrent (RNN) networks on the same footing. This is achieved by unrolling the RNN in time. The strategy is to treat learning as Bayesian inference and to write the network as a conditional density $P(f|\boldsymbol{x})$ that is an integral over collective fields encoding pre-activation covariances, integrate out all microscopic Gaussian parameters, and then take the large-width limit $N \to \infty$ via a saddle-point analysis. In this formulation the only structural difference between DNNs and RNNs is encoded by a simple masking operator on the time indices (equivalent to layers in the DNN case), while the scaling of the parameters and auxiliary fields with networks width $N$ remains identical across architectures.

### A.3.1. DERIVATION OF NETWORK PRIOR

We here derive the concrete form of the network prior (14) for the architecture (11). The computation follows closely previous works Segadlo et al. (2022); Fischer et al. (2024); Lauditi et al. (2025). We also give a compact overview of the derivation in Section A.7 that emphasizes the choice of scaling.

Since the cumulants of an observable (15) over the posterior over weights can be obtained equivalently from the network prior (14) by taking the logarithm and considering suitable derivatives, it is sufficient to derive the form of the prior. To this end, it is convenient to decompose the network prior into a chain of conditional probabilities

$$P(y|\boldsymbol{x}) = \int \mathcal{N}\big(y|f', \mathsf{K}\,\mathbb{I}\big)\, P(f'|\boldsymbol{x})\, df'\,, \tag{18}$$

with

$$P(f'|\boldsymbol{x}) = \langle \delta(f' - f(\boldsymbol{U}, \boldsymbol{V}, \boldsymbol{W}, \boldsymbol{x})) \rangle_{\boldsymbol{U} \sim \mathcal{N}_U, \boldsymbol{V} \sim \mathcal{N}_V, \boldsymbol{W} \sim \mathcal{N}_W}.$$

The latter can further be decomposed as

$$P(f'|\boldsymbol{x}) = \int P(f'|\boldsymbol{h})\, P(\boldsymbol{h}|\boldsymbol{x})\, d\mathbf{h}\,. \tag{19}$$

The Dirac constraint is rewritten in its Fourier representation as

$$\delta(f' - f) \equiv \prod_{p=1}^{P} \prod_{t=2}^{T} \delta(f_p'^t - f_p^t)$$

$$= \prod_{p=1}^{P} \prod_{t=2}^{T} \frac{1}{2\pi i} \int_{-i\infty}^{i\infty} d\tilde{f}_p^t \, \exp\Big\{ \tilde{f}_p^t \big[ f_p'^t - f_p^t \big] \Big\}\,,$$

which allows us to compute the expectation value over $\boldsymbol{V}$ as

$$P(f'|\boldsymbol{h}) = \int \mathcal{D}\tilde{f} \, \Big\langle \exp\Big\{ \sum_{t,p} \tilde{f}_p^t \big[ f_p'^t - \sum_i V_i \phi_{p,i}^{t-1} \big] \Big\} \Big\rangle_{\boldsymbol{V} \sim \mathcal{N}_V(0, v/N^2)}$$

$$= \exp\Big\{ \sum_{t,p} \tilde{f}_p^t f_p'^t + \frac{1}{2} \sum_{p,p';t,t'} \tilde{f}_p^t \tilde{f}_{p'}^{t'} \frac{v}{N^2} \sum_i \phi_{p,i}^{t-1} \phi_{p',i}^{t'-1} \Big\}\,,$$

where we define the integral measure $\int \mathcal{D}\tilde{f} = \Big( \prod_{p=1}^{P} \prod_{t=2}^{T} \frac{1}{2\pi i} \int_{-i\infty}^{i\infty} d\tilde{f}_p^t \Big)$. We write the latter expression for short as

$$P(f'|\boldsymbol{h}) = \int \mathcal{D}\tilde{f} \, \exp\Big\{ \sum_t \tilde{f}^{t\mathsf{T}} f'^t + \frac{1}{2} \sum_{t,t'} \tilde{f}^{t\mathsf{T}} \frac{v}{N^2} \big[ \boldsymbol{\phi}^{t-1} \cdot \boldsymbol{\phi}^{t'-1\mathsf{T}} \big] \tilde{f}^{t'} \Big\}\,, \tag{20}$$

where $\boldsymbol{a} \cdot \boldsymbol{b}$ denotes an inner product over $i = 1, \dots, N$ and $a^\mathsf{T} b$ an inner product over $p = 1, \dots, P$. We showed how we can write (18) as:

$$P(y|\boldsymbol{x}) = \int \Big[ \int \mathcal{N}\big(y|f', \mathsf{K}\,\mathbb{I}\big)\, P(f'|\boldsymbol{h})\, df' , \Big] P(\boldsymbol{h}|\boldsymbol{x})\, d\boldsymbol{h} \tag{21}$$

Performing the integral over $f'$, yields

$$P(y|\boldsymbol{h}) = \int \mathcal{N}\big(y|f', \mathsf{K}\,\mathbb{I}\big)\, P(f'|\boldsymbol{h})\, df' \tag{22}$$

$$= \int \mathcal{D}\tilde{f}\, \exp\Big\{\sum_t \tilde{f}^{t\mathsf{T}} y^t + \frac{1}{2}\sum_t \frac{\kappa}{N}\tilde{f}^{t\mathsf{T}}\tilde{f}^t + \frac{1}{2}\sum_{t,t'} \tilde{f}^{t\mathsf{T}} \frac{v}{N^2}\big[\boldsymbol{\phi}^{t-1}\cdot\boldsymbol{\phi}^{t'-1\mathsf{T}}\big]\,\tilde{f}^{t'}\Big\},$$

where we used the parameter $\mathsf{K} = \kappa/N$. We again use the Fourier representation of the delta function to enforce the evolution of the network dynamics (11) for each sample $p$ and each time $t$ as

$$P(\boldsymbol{h}|\boldsymbol{x}) = \prod_t \langle \delta(\boldsymbol{h}^t - \boldsymbol{W}^{(t-1)}\boldsymbol{\phi}^{t-1} - \boldsymbol{U}\boldsymbol{x}^{t-1})\rangle_{\boldsymbol{U}\sim\mathcal{N}_U,\,\{\boldsymbol{W}^{(t)}\}\sim\mathcal{N}_W} \tag{23}$$

$$= \int \mathcal{D}\tilde{\boldsymbol{h}}\, \exp\Big\{\sum_t \tilde{\boldsymbol{h}}^{t\mathsf{T}}\cdot\boldsymbol{h}^t + \frac{1}{2}\sum_{t,t'} \tilde{\boldsymbol{h}}^t\cdot\Big[\bar{\delta}^{t-1,t'-1}\frac{w}{N}\boldsymbol{\phi}^{t-1}\cdot\boldsymbol{\phi}^{\mathsf{T}t'-1} + \frac{u}{D}\boldsymbol{x}^{t-1}\cdot\boldsymbol{x}^{t'-1\mathsf{T}}\Big]\cdot\tilde{\boldsymbol{h}}^{t'}\Big\}.$$

Where we used the identity $\mathbb{E}_{w\sim\mathcal{N}(0,\Sigma)}\big[e^{a^\top w}\big] = \exp\big(\frac{1}{2}a^\top \Sigma a\big)$ and introduced the symbol

$$\bar{\delta}^{tt'} = \begin{cases} \delta^{tt'} & \text{DNN} \\ 1 & \text{RNN} \end{cases}.$$

The difference arises, because for the RNN there is only a single weight matrix $\boldsymbol{W}$ valid for all timesteps $t$, thus its entries are perfectly correlated across time, while in the DNN, the matrices $\boldsymbol{W}^{(t)}$ are drawn from prior distributions which are independent across different layers $t$.

One notes that due to the appearance of the inner products "$\cdot$" of the fields $\tilde{\boldsymbol{h}}$, the exponent factorizes across neuron indices $i$, which allows us to reduce the $N$ integrals over $\boldsymbol{h}$ and $\tilde{\boldsymbol{h}}$ to a single integral each. We introduce auxiliary fields for terms that contain an inner product over the neuron index. which we anticipate will concentrate in the large-$N$ limit, thus defining

$$\Phi^{tt'} := \frac{1}{N}\boldsymbol{\phi}^t\cdot\boldsymbol{\phi}^{t'\mathsf{T}} \in \mathbb{R}^{P\times P}\,, \tag{24}$$

$$\mathbb{X}^{tt'} := \frac{1}{D}\boldsymbol{x}^t\cdot\boldsymbol{x}^{t'\mathsf{T}} \in \mathbb{R}^{P\times P}\,.$$

Likewise, the kernels $\mathbb{X}^{tt'}_{pp'} := \frac{1}{D}\sum_i^D x^t_{p,i} x^{t'}_{p',i}$ and $\mathbb{Y}^{tt'}_{pp'} := y^t_p y^{t'}_{p'}$ denote input and label kernels, and $\mathbb{H}^{tt'}_{pp'} := \frac{1}{N}\sum_i^N h^t_{p,i} h^{t'}_{p',i}$ is the kernel for the pre-activations, defined in analogy to the kernel $\Phi$ for the post-activations. Enforcing the former definition by introducing a second auxiliary field $\tilde{\tilde{C}} \in i\,\mathbb{R}^{PT\times PT}$ one inserts a Dirac constraint in the form

$$\delta(-\Phi^{tt'} + \tfrac{1}{N}\boldsymbol{\phi}^t\cdot\boldsymbol{\phi}^{t'\mathsf{T}}) = \int \mathcal{D}\tilde{\tilde{C}}\, \exp\Big\{-\sum_{t,t';p,p'} \tilde{\tilde{C}}^{tt'}_{pp'}\Phi^{tt'}_{pp'} + \tilde{\tilde{C}}^{t,t'}_{pp'}\,\tfrac{1}{N}\boldsymbol{\phi}^t_p\cdot\boldsymbol{\phi}^{t'}_{p'}\Big\}.$$

where we used $\int \mathcal{D}\tilde{\tilde{C}} = \Big(\prod_{p,p'=1}^P \prod_{t,t'=1}^{T_-} \frac{1}{2\pi i}\int_{-i\infty}^{i\infty} d\tilde{\tilde{C}}^{tt'}_{pp'}\Big)$. This allows us to use equation (20) to write as

$$P(y|\Phi,\boldsymbol{x}) = \int \mathcal{D}f\, \exp\Big\{f^\mathsf{T} y + \frac{1}{2}\sum_{t,t'} f^{t\mathsf{T}}\big(\frac{v}{N}\Phi^{t-1,t'-1} + \frac{\kappa}{N}\big)f^{t'}\Big\},$$

$$= \exp\Big\{-\frac{N}{2}\sum_{tt'} y^{t\mathsf{T}}\big((v\Phi^- + \kappa)^{-1}\big)^{tt'} y^{t'} - \frac{1}{2}\ln|v\Phi^- + \kappa| + \frac{PT_-}{2}\ln N\Big\},$$

where we performed the Gaussian integral over $\tilde{f}$ in (22) and introduced the shorthand $\Phi^-$ as the one time-step shifted version of the original matrix, i.e. $(\circ^-)^{tt'} := \circ^{t-1,t'-1}$. Inserting (23) and (23) in (21), and inserting the definitions of the auxiliary fields (24) we obtain .

$$P(y|\boldsymbol{x}) \propto \int \mathcal{D}\Phi \exp\left\{-\frac{N}{2}\sum_{tt'} y^{t\mathsf{T}}((v\Phi^- + \kappa)^{-1})^{tt'} y^{t'} + \mathcal{O}(1)\right\} \tag{25}$$
$$\times \int \mathcal{D}\tilde{\tilde{C}} \exp\left\{-\sum_{tt'} \tilde{\tilde{C}}^{tt'\mathsf{T}}\Phi^{tt'} + N\,\mathcal{W}(\tilde{\tilde{C}}/N|\Phi,\mathbb{X})\right\},$$

$$\mathcal{W}(\tilde{C}|\Phi,\mathbb{X}) := \ln \int \mathcal{D}\tilde{h}\, \exp\left\{\sum_{t,t'}\phi^{t\mathsf{T}}\tilde{C}^{t,t'}\phi^{t'} + \sum_t \tilde{h}^{t\mathsf{T}}\tilde{h}^t + \frac{1}{2}\sum_{t,t'}\tilde{h}^{t\mathsf{T}}\left[w\bar{\delta}^{tt'}\Phi^{t-1,t'-1} + u\mathbb{X}^{t-1,t'-1}\right]\tilde{h}^{t'}\right\},$$

where we used that the integrals over $\boldsymbol{h}$ and over $\tilde{\boldsymbol{h}}$ factorize over neurons $i = 1, \ldots, N$ conditional on the introduced order parameter $\Phi$, and thus yield the same integral to the $N$th power – hence the factor $N\,\mathcal{W}$ appearing and we dropped the term $\frac{1}{2}\ln|v\Phi^- + \kappa|$ in the first line that scales as order unity on $N$ as well as constant terms. A short way of writing the latter line is as

$$\mathcal{W}(\tilde{C}|\Phi,\mathbb{X}) := \ln \left\langle \exp\left\{\sum_{t,t'}\sum_{p,p'}\tilde{C}^{tt'}_{pp'}\phi(h^t_p)\phi(h^{t'}_{p'})\right\}\right\rangle_{h\sim\mathcal{N}(0,\,w\bar{\delta}^{tt'}\Phi^{t-1,t'-1}+u\mathbb{X}^{t-1,t'-1})}, \tag{26}$$

which also corresponds to taking the integral over the $\tilde{h}$ fields.

We note that $N\,\mathcal{W}(\tilde{\tilde{C}}/N|\Phi,\mathbb{X})$ has the form of a cumulant-generating function for the random variable $\Phi$ and hence $P(\Phi|\mathbb{X}) = \int \mathcal{D}\tilde{\tilde{C}}\,\exp\left(-\sum_{tt'}\tilde{\tilde{C}}^{tt'\mathsf{T}}\Phi^{tt'} + N\,\mathcal{W}(\tilde{\tilde{C}}/N|\Phi,\mathbb{X})\right)$ is the Fourier representation of the probability distribution $P(\Phi|\mathbb{X})$. The trailing factor $N$ and the factor $N^{-1}$ coming together with the source variable come in what is known as the scaling form – they indicate that first derivatives are $\mathcal{O}(1)$, while all higher derivatives are suppressed with at least $N^{-1}$, indicating that the mean dominates the distribution of $\Phi$.

An alternative way of seeing this is to define the rescaled, intensive field $\tilde{C} := \tilde{\tilde{C}}/N$ which then leads to

$$P(\Phi|\mathbb{X}) \propto \int \mathcal{D}\tilde{C} \exp\left\{-N\sum_{tt'}\tilde{C}^{tt'\mathsf{T}}\Phi^{tt'} + N\,\mathcal{W}(\tilde{C}|\Phi,\mathbb{X})\right\}$$
$$\stackrel{N\to\infty}{\simeq} \exp\left\{N\sup_{\tilde{C}}\left[-\sum_{tt'}\tilde{C}^{tt'\mathsf{T}}\Phi^{tt'} + \mathcal{W}(\tilde{C}|\Phi,\mathbb{X})\right]\right\}$$
$$=: \exp\left\{-N\,\Gamma(\Phi|\mathbb{X})\right\}. \tag{27}$$

The latter expression follows from a saddle point approximation of the Fourier integral over $\tilde{C}$ and shows that a rate function $\Gamma$ appears in the exponent. We note that the entire integrand appearing in (25) has a trailing factor $N$, so that we may also take the integral over $\Phi$ in saddle point approximation, which leads to

$$\ln P(y|\boldsymbol{x})/N \simeq \sup_{\Phi} S(\Phi|\mathbb{X},\mathbb{Y}), \quad S(\Phi|\mathbb{X},\mathbb{Y}) := \left[-\frac{1}{2}\sum_{tt'}y^{t\mathsf{T}}((v\Phi^- + \kappa)^{-1})^{tt'}y^{t'} - \Gamma(\Phi|\mathbb{X})\right]. \tag{28}$$

This object is our main theoretical result, which recasts the problem of learning into a variational problem of determining the maximum of the right hand side with regard to $\Phi$.

### A.3.2. SADDLE-POINT EQUATIONS FOR THE RNN CASE

We now instantiate the RNN case by replacing $\bar{\delta}^{tt'} = 1$ and redefine contractions $a^\mathsf{T}b := \sum_{tp}a^t_p b^t_p$ to run over patters and time (unlike only patterns as before) to ease notation. The supremum condition (saddle point equation) for $\tilde{C}$ in (27) then becomes

$$\Phi_{pp'}^{tt'} = \frac{\partial}{\partial \tilde{C}_{pp'}^{tt'}} \mathcal{W}(\tilde{C}|\Phi, \mathbb{X}) = \frac{1}{Z} \langle \phi(h_p^t)\phi(h_{p'}^{t'}) \exp\{\sum_{pp'}\sum_{tt'} \phi(h_p^t)\tilde{C}_{pp'}^{tt'}\phi(h_{p'}^{t'})\}\rangle_{h\sim\mathcal{N}(0, \, w\Phi^- + u\mathbb{X}^-)}$$

$$=: \langle \phi(h_p^t)\phi(h_{p'}^{t'})\rangle_{P(h|\boldsymbol{x}, y)},$$

where $Z = \langle \exp\{\sum_{pp'}\sum_{tt'} \phi(h_p^t)\tilde{C}_{pp'}^{tt'}\phi(h_{p'}^{t'})\}\rangle_{h\sim\mathcal{N}(0, \, w\Phi^- + u\mathbb{X}^-)}$ is a the normalization constant. The expectation thus is taken with respect to the posterior $P(h|\boldsymbol{x}, y) = P(h|\Phi, \tilde{C}, \mathbb{X}, \mathbb{Y})$, i.e. the kernels $\Phi, \tilde{C}, \mathbb{X}, \mathbb{Y}$ are sufficient statistics for this measure.

To close these equations, we still need an expression for $\tilde{C}$. For that we re-write the action containing the label term $S(\Phi|\mathbb{X}, \mathbb{Y}) = -\frac{1}{2}\text{tr}\big(\mathbb{Y}(v\Phi^- + \kappa)^{-1}\big) - \Gamma(\Phi|\mathbb{X})$ and the cumulant generating function (26)

$$\mathcal{W}(\tilde{C}|\Phi, \mathbb{X}) := \ln \int dh \, \exp\left( -\frac{1}{2}h^\mathsf{T}(w[\![\Phi^-]\!] + u\mathbb{X}^-)^{-1}h + \phi^\mathsf{T}\tilde{C}\phi\right) - \frac{1}{2}\ln|w[\![\Phi^-]\!] + u\mathbb{X}^-|,$$

in term of kernel matrices. We had introduced the kernel $\mathbb{H}_{pp'}^{tt'} := \frac{1}{N}\sum_i^N h_{p,i}^t h_{p',i}^{t'}$, which we can now write as $\mathbb{H} := \langle hh^\mathsf{T}\rangle_{P(h|\boldsymbol{x}, y)}$. Using $\frac{\partial}{\partial \Phi^-}\ln|w[\![\Phi^-]\!] + u\mathbb{X}^-| = w(w[\![\Phi^-]\!] + u\mathbb{X}^-)^{-1}$ and demanding stationarity of $S$ (28) with respect to $\Phi$ we obtain:

$$0 \overset{!}{=} \frac{\partial}{\partial\Phi}S(\Phi|\mathbb{X}, \mathbb{Y}) \Leftrightarrow \tilde{C} = \frac{1}{2}v(v\Phi^- + \kappa)^{-1}\begin{pmatrix} \mathbb{Y} & - & 0 \end{pmatrix}(v\Phi^- + \kappa)^{-1}$$
$$+ \frac{1}{2}w(w[\![\Phi^-]\!] + u\mathbb{X}^-)^{-1}\begin{pmatrix} \mathbb{H} & - & (w[\![\Phi^-]\!] + u\mathbb{X}^-) \end{pmatrix}(w[\![\Phi^-]\!] + u\mathbb{X}^-)^{-1}.$$
$$(29)$$

This derivation yields the first main result of this work, cf. (3).

### A.3.3. EFFECT OF PARTIAL TEMPORAL SUPERVISION

In the derivation above we assumed that the observation noise variance $\kappa$ is the same for all timesteps. We now generalize to partial temporal supervision, where only a subset of timesteps is observed. The key insight is probabilistic: since $\kappa$ controls the observation noise variance, setting $\kappa_t \to \infty$ at unobserved timesteps $t$ makes the corresponding labels infinitely uncertain, hence uninformative. Therefore, the label-dependent term in (29) should vanish at these points.

To verify this formally, let $\bar{\mathcal{T}}$ denote the set of unobserved timesteps and define the projector $\mathbf{U}\mathbf{U}^\mathsf{T} = \sum_{t\in\bar{\mathcal{T}}} \mathbf{e}_t\mathbf{e}_t^\mathsf{T}$ onto this subspace, where $\{\mathbf{e}_t\}_{t=1\ldots T}$ is the standard basis. The slack matrix becomes $\boldsymbol{\kappa} = \kappa(\mathbf{I}_T - \mathbf{U}\mathbf{U}^\mathsf{T}) + \kappa_\infty \mathbf{U}\mathbf{U}^\mathsf{T}$. For any matrix $\mathbf{A}$ like the ones appearing in (29), the Woodbury identity gives

$$(\mathbf{A} + \kappa_\infty\mathbf{U}\mathbf{U}^\mathsf{T})^{-1} = \mathbf{A}^{-1} - \mathbf{A}^{-1}\mathbf{U}(\kappa_\infty^{-1} + \mathbf{U}^\mathsf{T}\mathbf{A}^{-1}\mathbf{U})^{-1}\mathbf{U}^\mathsf{T}\mathbf{A}^{-1}$$
$$\overset{\kappa_\infty\to\infty}{\longrightarrow} \mathbf{A}^{-1} - \mathbf{A}^{-1}\mathbf{U}(\mathbf{U}^\mathsf{T}\mathbf{A}^{-1}\mathbf{U})^{-1}\mathbf{U}^\mathsf{T}\mathbf{A}^{-1}.$$

This expression has vanishing support on $\text{span}\{\mathbf{e}_t\}_{t\in\bar{\mathcal{T}}}$, as can be verified by multiplying with $\mathbf{U}$ from the left and $\mathbf{U}^\mathsf{T}$ from the right. Thus the immediate 'force' in the first line of (29) due to labels vanishes at unobserved timesteps, and only the NNGP prior (second line) remains. Note that due to indirect feedback through observed points, the posterior $\langle hh^\mathsf{T}\rangle$ will still deviate from the NNGP prior; see Fischer et al. (2024).

### A.3.4. REDUCTION TO DNN CASE

In their standard formulation, DNNs differ from the general architecture developed in Section A.3.1 in three aspects:

1. There is only input in the first layer, $\boldsymbol{x}^t = \delta^{t0}\boldsymbol{x}^0$.

2. There is only output supervision on the last layer. This does not amount to just setting $y^{t<T} = 0$, instead, we loosen the slack in all but the last layer. This can also be interpreted as believing that hypothetical evidence $y^{t<T}$ is non-salient (i.e., measurements that have been corrupted by uninformative noise $\xi^{t<T} \sim \mathcal{N}(0, \kappa_\infty\mathbf{I}^{t<T}), \kappa_\infty \gg \kappa$).

3. The weights under the prior are pairwise independent random variables across layers, $P(\boldsymbol{W}^{(t)}, \boldsymbol{W}^{(t+1)}) = \mathcal{N}_{\boldsymbol{W}}(\boldsymbol{W}^{(t)})\mathcal{N}_{\boldsymbol{W}}(\boldsymbol{W}^{(t+1)})$. After marginalization, this amounts to masking all off-diagonal elements to be zero in the relevant expressions, denoted as $[\![\Phi]\!] \to \mathrm{diag}(\Phi)$ (the vectorized version of replacing $\bar{\delta}^{tt'} \to \delta^{tt'}$), i.e. only sampling the measure in the covariance that is the diagonal of $\Phi$.

This leads to the following effect on the saddle-point equations. The kernel order parameter reads

$$\Phi^{tt'} = \frac{\partial \mathcal{W}}{\partial \tilde{C}^{tt'}}(\Phi; \tilde{C}) = \langle \phi(h^t)\phi(h^{t'})\rangle_{P(h|\Phi,\tilde{C},\mathbb{Y},\mathbb{X})} \quad . \tag{30}$$

The supremum condition for $\tilde{C}$ in (27) becomes

$$0 \overset{!}{=} \frac{\partial S}{\partial \Phi} \quad \Leftrightarrow \quad \tilde{C} = \begin{aligned} &\tfrac{1}{2}v\big(v\Phi^- + \kappa\big)^{-1}\big( \quad \mathbb{Y} \quad - \quad 0 \quad \big)\big(v\Phi^- + \kappa\big)^{-1}\\ &+\tfrac{1}{2}w\big(w\,\mathrm{diag}(\Phi^-) + u\mathbb{X}^-\big)^{-1}\big( \quad \mathbb{H} \quad - \quad (w\,\mathrm{diag}(\Phi^-) + u\mathbb{X}^-) \quad \big)\big(w\,\mathrm{diag}(\Phi^-) + u\mathbb{X}^-\big)^{-1}. \end{aligned} \tag{31}$$

The first line can be interpreted as a tilt $\tilde{C}_y$ due to the ($\kappa$-regularized) constraint to match the labels $y$, and the second line as a tilt $\tilde{C}_h$ due to the definition of the network's forward pass.

Due to the diagonal-only coupling, this set of equations can be reduced. To this end, we make the diagonal Ansatz $\Phi = \Phi$, $\tilde{C} = \tilde{C}$, $\mathbb{H} = \mathbb{H}$. Then, the conjugate saddle point equation (30) becomes

$$\begin{aligned} \tilde{C}^{TT} &= \tfrac{1}{2} && v\big(v\Phi^{T\text{-}T\text{-}} + \kappa\big)^{-1}\big( && \mathbb{Y}^{TT} && - && 0 && \big)\big(v\Phi^{T\text{-}T\text{-}} + \kappa\big)^{-1}, \\ \tilde{C}^{tt} &= && \tfrac{1}{2}w\big(w\Phi^{t-1,t-1}\big)^{-1}\big( && \langle h^t h^{t\,\mathsf{T}}\rangle && - && w\Phi^{t-1,t-1} && \big)\big(w\Phi^{t-1,t-1}\big)^{-1}, \quad 2 \le t < T-1 \\ \tilde{C}^{11} &= && \tfrac{1}{2}w\big(u\mathbb{X}^{0,0}\big)^{-1}\big( && \langle h^1 h^{1\,\mathsf{T}}\rangle && - && u\mathbb{X}^{0,0} && \big)\big(u\mathbb{X}^{0,0}\big)^{-1}, \end{aligned} \tag{32}$$

where we used $\kappa^{\le T} = \kappa_\infty \to \infty$. (32) reproduces the saddle-point equations previously derived for DNNs Fischer et al. (2024); Lauditi et al. (2025).

## A.4. Special case of linear recurrent networks

We here consider the special case of a linear activation function $\phi(h) = h$, in which the effective probability and saddle point equations simplify considerably.

### A.4.1. KERNEL MEAN-FIELD THEORY FOR THE LINEAR CASE (WITHOUT LABEL TERM)

We derived the rate function (27) and the cumulant-generating function (26) for the linear case, first for the case without the label term. In that case we get

$$P(\mathbb{H}|\mathbb{X}) \propto \exp\{-N\Gamma(\mathbb{H}|\mathbb{X})\}, \quad \Gamma(\mathbb{H}|\mathbb{X})) = \sup_{\tilde{C}}\left(\mathrm{tr}\,\tilde{C}^{\mathsf{T}}\mathbb{H} - \mathcal{W}(\tilde{C}|\mathbb{X})\right),$$

$$\mathcal{W}(\tilde{C}|\mathbb{X}) = \ln\int dh\,\exp\left\{-\tfrac{1}{2}h^{\mathsf{T}}\big(w\mathbb{H}^- + u\mathbb{X}^-\big)^{-1}h + h^{\mathsf{T}}\tilde{C}\,h - \tfrac{1}{2}N\ln\big|w\mathbb{H}^- + u\mathbb{X}\big|\right\}.$$

Using the supremum condition for $\tilde{C}$ in (27)

$$0 \overset{!}{=} \frac{d}{d\tilde{C}}\left(\tilde{C}^{\mathsf{T}}\mathbb{H} - \mathcal{W}(\tilde{C}|\mathbb{X})\right) \Leftrightarrow \tilde{C}(\mathbb{H}) = \frac{1}{2}\big(w\mathbb{H}^- + u\mathbb{X}^-\big)^{-1} - \frac{1}{2}\mathbb{H}^{-1},$$

we get by plugging in

$$\begin{aligned}
\Gamma(\mathbb{H}|\mathbb{X}) &= \operatorname{tr}\tilde{C}(\mathbb{H})^{\mathsf{T}}\mathbb{H} - \mathcal{W}(\tilde{C}(\mathbb{H})|\mathbb{X})\\
&= \tfrac{1}{2}\operatorname{tr}\left[\mathbb{H}\left((w\mathbb{H}^- + u\mathbb{X}^-)^{-1} - \mathbb{H}^{-1}\right)\right] \quad -\ln\int dh\,\exp\Big\{ -\tfrac{1}{2}h^{\mathsf{T}}(w\mathbb{H}^- + u\mathbb{X}^-)^{-1}h \qquad +h^{\mathsf{T}}\tilde{C}h\\
&= \tfrac{1}{2}\operatorname{tr}\left[\mathbb{H}(w\mathbb{H}^- + u\mathbb{X}^-)^{-1} - \mathbb{I}\right]\quad -\ln\int dh\,\exp\Big\{ -\tfrac{1}{2}h^{\mathsf{T}}(w\mathbb{H}^- + u\mathbb{X}^-)^{-1}h \;+\tfrac{1}{2}h^{\mathsf{T}}\left((w\mathbb{H}^- + u\mathbb{X}^-)^{-1} - \mathbb{H}^{-1}\right)h\\
&= \tfrac{1}{2}\operatorname{tr}\left[\mathbb{H}(w\mathbb{H}^- + u\mathbb{X}^-)^{-1} - \mathbb{I}\right]\quad -\ln\int dh\,\exp\Big\{ \qquad -\tfrac{1}{2}h^{\mathsf{T}}\mathbb{H}^{-1}h\\
&\overset{+\text{const.}}{=} \tfrac{1}{2}\operatorname{tr}\left[\mathbb{H}(w\mathbb{H}^- + u\mathbb{X}^-)^{-1}\right]
\end{aligned}$$

(33)

where we note that $\tfrac{1}{2}\operatorname{tr}\left[\mathbb{H}(w\mathbb{H}^- + u\mathbb{X}^-)^{-1}\right] - \tfrac{1}{2}\ln\frac{|\mathbb{H}|}{|w\mathbb{H}^- + u\mathbb{X}^-|} = D_{\mathrm{KL}}\left(\mathcal{N}_h(0,\mathbb{H})\,\|\,\mathcal{N}_h(0,w[\![\mathbb{H}^-]\!] + u\mathbb{X}^-)\right)$ is the Kullback-Leibler divergence between two Gaussians in $h \in \mathbb{R}^T$. Note that this result can also be obtained directly from noticing that the supremum condition on $\tilde{C}$ is made precisely such that $\mathbb{H} = \langle hh^{\mathsf{T}}\rangle$ (by definition of the Legendre transform). Hence the integral over $h$ is a Gaussian with covariance $\mathbb{H}$.

### A.4.2. CLOSED-FORM SOLUTION (WITH LABEL TERM)

Similarly to previous work, we can devise a closed-form solution for the kernels by eliminating $\tilde{C}$, accounting for the tilting through the label term $\mathbb{Y}$ in $S(\Phi|\mathbb{X}, \mathbb{Y})$. The basic reason this is possible is that per (31)

$$\tilde{C} = \frac{1}{2}w(w[\![\mathbb{H}]\!] + u\mathbb{X})^{-1}\left([hh^{\mathsf{T}}]^+_{\Phi,\tilde{C}^+} - (w[\![\mathbb{H}]\!] + u\mathbb{X})\right)(w[\![\mathbb{H}]\!] + u\mathbb{X})^{-1} \quad + \quad \frac{1}{2}v(v\mathbb{H} + \kappa)^{-1}\mathbb{Y}^+(v\mathbb{H} + \kappa)^{-1},$$

and we in turn know that at the saddle it holds that $\Phi^+ = [hh^{\mathsf{T}}]^+_{\Phi,\tilde{C}^+}$ giving

$$\tilde{C} = \frac{1}{2}w(w[\![\mathbb{H}]\!] + u\mathbb{X})^{-1}\left(\mathbb{H}^+ - (w[\![\mathbb{H}]\!] + u\mathbb{X})\right)(w[\![\mathbb{H}]\!] + u\mathbb{X})^{-1} \quad + \quad \frac{1}{2}v(v\mathbb{H} + \kappa)^{-1}\mathbb{Y}^+(v\mathbb{H} + \kappa)^{-1}.$$

Finally, for the linear case we can read-off $\Phi$ directly as the coefficient of $h$ in the action, giving $(w[\![\mathbb{H}^-]\!] + u\mathbb{X}^-)^{-1} - 2\tilde{C} = \mathbb{H}^{-1}$. This gives a **closed-form relation** for $\mathbb{H}$

$$(w[\![\mathbb{H}^-]\!] + u\mathbb{X}^-)^{-1} - \mathbb{H}^{-1} = w(w[\![\mathbb{H}]\!] + u\mathbb{X})^{-1}\left(\mathbb{H}^+ - (w[\![\mathbb{H}]\!] + u\mathbb{X})\right)(w[\![\mathbb{H}]\!] + u\mathbb{X})^{-1} \tag{34}$$
$$+ v(v\mathbb{H} + \kappa)^{-1}\mathbb{Y}^+(v\mathbb{H} + \kappa)^{-1}$$

where the last layer's term $2\tilde{C}_h^{TT} = \mathbb{H}^{TT} - w\Phi^{T\text{-}T\text{-}} = 0$ because the last layer is free to vary marginally.

This result of course corresponds to the stationary point of (28) in the special case of using (33)

$$S(\mathbb{H}|\mathbb{Y}, \mathbb{X}) = -\frac{1}{2}\operatorname{tr}\left[\mathbb{Y}(v\mathbb{H}^- + \kappa)^{-1}\right] - \frac{1}{2}\operatorname{tr}\left[\mathbb{H}(w\mathbb{H}^- + u\mathbb{X}^-)^{-1}\right] + \frac{1}{2}\ln\frac{|\mathbb{H}|}{|w\mathbb{H}^- + u\mathbb{X}^-|}.$$

### A.4.3. PERTURBATIVE SOLUTION

We here derive the perturbative to solution to (34) that we used in Section 3.3 in the main text. For the NNGP solution (i.e., absence of learning signal $\mathbb{Y}$), we have $\mathbb{H}_0 = w[\![\mathbb{H}_0^-]\!] + u\mathbb{X}^-$. We then make a perturbation expansion $\mathbb{H} = \mathbb{H}_0 + \Delta$, $\Delta = \Delta_1 + \Delta_2 + \ldots$, where every order groups powers of $\mathbb{Y}$. We perform this expansion up to second order, which is the minimal order at which the architecture-dependent effects that we are interested in appear. Thus, we insert the perturbative ansatz into (34) to get the expression

$$(w(\mathbb{H}_0+\Delta)^- + u\mathbb{X}^-)^{-1} - (\mathbb{H}_0+\Delta)^{-1} = w\left(w[\![\mathbb{H}_0+\Delta]\!] + u\mathbb{X}\right)^{-1}(\Delta^+ - w[\![\Delta]\!])\left(w[\![\mathbb{H}_0+\Delta]\!] + u\mathbb{X}\right)^{-1} + v\left(v(\mathbb{H}_0+\Delta)+\kappa\right)^{-1}\mathbb{Y}^+\left(v(\mathbb{H}_0+\Delta)+\kappa\right)^{-1}$$

and group terms in zeroth, first, and second order in $\mathbb{Y}$ to compare coefficients.

**Zeroth order**   The terms of zeroth order

$$\left(w\mathbb{H}_0^- + u\mathbb{X}^-\right)^{-1} - \mathbb{H}_0^{-1} = 0$$

are fulfilled by having chosen the NNGP solution as the expansion point, for which $[\![\mathbb{H}_0]\!] = \mathbb{H}_0$.

**First order**   To ease notation, we introduce the NNGP "propagators" $\mathbb{G}_h := \left(w\mathbb{H}_0^- + u\mathbb{X}^-\right)^{-1} \equiv (\mathbb{H}_0)^{-1}$, $\mathbb{G}_h^+ := \left(w\mathbb{H}_0 + u\mathbb{X}\right)^{-1}$, $\mathbb{G}_y := \left(v\mathbb{H}_0 + \kappa\right)^{-1}$. Expanding the LHS resolvents and the RHS linear terms yields the defining equation for $\triangle_1$:

$$-(\mathbb{G}_h w [\![\triangle_1^-]\!] \mathbb{G}_h - \mathbb{G}_h \triangle_1 \mathbb{G}_h) = w\mathbb{G}_h^+ \left(\triangle_1^+ - w[\![\triangle_1]\!]\right) \mathbb{G}_h^+ + v\mathbb{G}_y \mathbb{Y}^+ \mathbb{G}_y\,,$$

where we used $[\![\mathbb{H}_0]\!] = \mathbb{H}_0$ because the NNGP solution is diagonal Segadlo et al. (2022). This moreover reveals that the first order change is proportional to $\mathbb{Y}$.

**Second order**   Grouping second-order terms $(\triangle_1)^2, \triangle_2$, we get

$$\begin{aligned}
-(\mathbb{G}_h w [\![\triangle_2^-]\!] \mathbb{G}_h - \mathbb{G}_h \triangle_2 \mathbb{G}_h) = &\qquad\qquad w\mathbb{G}_h^+ (\triangle_2^+ - w[\![\triangle_2]\!])\mathbb{G}_h^+ \\
+\mathbb{G}_h w [\![\triangle_1^-]\!] \mathbb{G}_h w [\![\triangle_1^-]\!] \mathbb{G}_h - \mathbb{G}_h \triangle_1 \mathbb{G}_h \triangle_1 \mathbb{G}_h &\quad -w\left(\mathbb{G}_h^+ w [\![\triangle_1]\!] \mathbb{G}_h^+ (\triangle_1^+ - w[\![\triangle_1]\!])\mathbb{G}_h^+ + (\sim)^\mathsf{T}\right) \\
&\qquad\qquad -v\left(\mathbb{G}_y v\triangle_1 \mathbb{G}_y \mathbb{Y}^+ \mathbb{G}_y + (\sim)^\mathsf{T}\right),
\end{aligned}$$

where $(\sim)^\mathsf{T}$ denotes the transpose of the preceding term. Notably, in the second line, masked $[\![\triangle_1]\!]$ and unmasked $\triangle_1$ terms appear in a product, giving rise to the contractions in the main text that allow for signal propagation.

### A.4.4. SOLUTION DEGENERACY IN LINEAR RNNS

Within the temporally-coherent phase of endpoint-supervised tasks, there can technically also be other solutions, for example where the hidden layer activity flips sign from one timestep to the next, $\boldsymbol{h}^{t+1} = -\boldsymbol{h}^t$. Such solutions are unstable, since they would be suppressed by an infinitesimal residual pathway (memory term) in (1), $\boldsymbol{h}^{t+1} = \ldots + (1-\alpha)\boldsymbol{h}^t$. To exclude this solution, we start learning from a small $\alpha$, which we subsequently anneal to zero.

### A.5. Analytics for temporally coherent phase in 4-layer linear RNNs in terms of label strength

**Setup**   We consider the case $T = 4$ with $\kappa = 0$. Let $\mathbb{H}_{0:4} \in \mathbb{R}^{4\times 4}$ be the kernel and $\mathbb{X}_{0:4}^- \in \mathbb{R}^{4\times 4}$ encode input correlations. Here, we are adopted Python-like slicing conventions that include the first and exclude the last index of a slice, i.e. $1{:}4 := (1,2,3)$, and will put temporal indices into *sub*scripts to avoid confusion with exponents. We work in the endpoint-supervised setting where in addition input is only supplied to the first layer, so

$$\mathbb{X}^- = \mathrm{diag}(1,0,0,0) = \mathbf{e}_0 \mathbf{e}_0^\mathsf{T}. \tag{35}$$

Moreover, we consider a single training sample, i.e. $P = 1$.

The log-probability (effective action) from (28) using (33) for the end-point supervised task with label $y_4$ can be written as

$$\ell(\mathbb{H}) := \ln P(\mathbb{H}|y, \boldsymbol{x})/N = -\frac{1}{2}\left[y_4^2\left(v\mathbb{H}^{T\text{-}T\text{-}} + \kappa\right)^{-1}\right] - \frac{1}{2}\mathrm{tr}[\mathbb{H}(\Sigma_h(\mathbb{H}))^{-1}] + \frac{1}{2}\ln\frac{|\mathbb{H}|}{|\Sigma_h(\mathbb{H})|}, \tag{36}$$

where we introduced the shorthand $\Sigma_h(\mathbb{H}) = w\mathbb{H}^- + u\mathbb{X}^-$ and used that, due to the supervision of the end point only, the probability of the label $y_4$ is with regard to the marginal distribution in the last layer alone (this is consistent with the consideration in Section A.3.3, sending the regularization noise $\kappa \to \infty$ for all unobserved timesteps). We will use the dimensionless control parameter

$$\lambda := \frac{y_4^2}{u\,w^2\,v}. \tag{37}$$

**Parameterization of the relevant $3 \times 3$ block**   In the $T = 4$ case, the nontrivial dependence of $\ell$ on off-diagonal elements is confined to the lower-right $3 \times 3$ block of $\mathbb{H}$, corresponding to times $1, 2, 3$. We denote this block by $\mathbb{H}_{1:4} \in \mathbb{R}^{3 \times 3}$ and explicitly parametrize it as

$$\mathbb{H}_{1:4} = \begin{bmatrix} a & b_2 & b_3 \\ b_2 & c & d \\ b_3 & d & e \end{bmatrix}, \quad \mathbb{H}_{1:3} := \begin{bmatrix} a & b_2 \\ b_2 & c \end{bmatrix}, \quad \mathbb{H}_{2:4} = \begin{bmatrix} c & d \\ d & e \end{bmatrix}.$$

The observed output at the final time is $y_4$, so that $v\,\mathbb{H}_{3,3} = ve$.

For the reduced $3 \times 3$ problem, the latent covariance over the last three times decomposes into a driven $1 \times 1$ block and an interior $2 \times 2$ block. The only architecture-dependent difference (RNN vs. DNN) is whether the interior block inherits the off-diagonal $b_2$:

$$\Sigma_h(\mathbb{H}) = \begin{cases} u\,\mathbf{e}_0\mathbf{e}_0^\intercal \ \oplus \ w\,\mathbb{H}_{1:3} = \begin{bmatrix} u & 0 & 0 \\ 0 & wa & wb_2 \\ 0 & wb_2 & wc \end{bmatrix} & \text{RNN (unmasked)} \\[4mm] u\,\mathbf{e}_0\mathbf{e}_0^\intercal \ \oplus \ w\,\text{diag}(\mathbb{H}_{1:3}) = \begin{bmatrix} u & 0 & 0 \\ 0 & wa & 0 \\ 0 & 0 & wc \end{bmatrix} & \text{DNN (masked)} \end{cases}, \tag{38}$$

where we used the input kernel (35) and employed $\mathbf{e}_0 = (1, 0, 0)^\intercal$ in the reduced 3-dimensional indexing.

**Diagonal first-order conditions at $b_2 = b_3 = d = 0$**   On the purely diagonal ansatz $(b_2, b_3, d) = (0, 0, 0)$, the RNN and DNN coincide. The saddle point conditions $\frac{\partial}{\partial \{a, c\}} \ell\big(\mathbb{H}(a, c, e)\big) \overset{!}{=} 0$ then imply

$$a^2 = \frac{u}{w}\,c, \qquad e = \frac{c^2}{a}. \tag{39}$$

We denote these stationarity conditions as "first-order-conditions" (FOC) in the following. The $e$–FOC $\frac{\partial}{\partial e} \ell\big(\mathbb{H}(a, c, e)\big) \overset{!}{=} 0$ reduces to a scalar relation between $e$, $c$ and the label:

$$\frac{1}{e} - \frac{1}{w\,c} + \frac{y_4^2}{v\,e^2} = 0. \tag{40}$$

These diagonal relations will later be used to locate the critical point in terms of $\lambda$.

**Architecture-dependent gradient and linear response of interior off-diagonals**   The first-order condition with respect to the interior off-diagonal $b_2$ differs between the RNN and DNN because $\Sigma_h(\mathbb{H})$ in (38) depends on $b_2$ only for the RNN. Writing the derivative by the matrix $\Sigma_h$ as

$$\frac{\partial \ell}{\partial \Sigma_h} = \frac{1}{2}\,\Sigma_h^{-1}(\mathbb{H} - \Sigma_h)\Sigma_h^{-1},$$

the chain-rule gives

$$\partial_{b_2}\ell = (\mathbb{H}^{-1})_{12} \ + \ \begin{cases} w\big(\Sigma_h^{-1}(\mathbb{H} - \Sigma_h)\Sigma_h^{-1}\big)_{23}, & \text{RNN,} \\ 0, & \text{DNN.} \end{cases}$$

Here we used two properties: first, that the first term of $\ell(\mathbb{H})$ does not depend on $b_2$, because it is $-\frac{1}{2}\left[ y_4^2\,\big(v\mathbb{H}^{\intercal\text{-}\intercal\text{-}} + \kappa\big)^{-1} \right]$, which only depends on $e$. Second, the trace can be explicitly calculated as $\text{tr}[\mathbb{H}(\Sigma_h(\mathbb{H}))^{-1}] = \text{sum}\Big( \begin{bmatrix} a & b_2 & b_3 \\ b_2 & c & d \\ b_3 & d & e \end{bmatrix} \odot \begin{bmatrix} 1/u & 0 & 0 \\ 0 & wc/|\mathbb{H}_{1:3}| & -wb_2/|\mathbb{H}_{1:3}| \\ 0 & -wb_2/|\mathbb{H}_{1:3}| & wa/|\mathbb{H}_{1:3}| \end{bmatrix} \Big)$, where we leveraged a block-inversion formula for $(\Sigma_h(\mathbb{H}))^{-1}$ and used $\text{tr}[AB] = \sum_{ij} A_{ij} B_{ij} =: \text{sum}(A \odot B)$; thus the $b_2$ dependence through $\mathbb{H}$ in the first term vanishes.

Thus only in the RNN is there an additional feedback term driven by the mismatch $\mathbb{H} - \Sigma_h$.

To analyze the onset of an interior off-diagonal $d$ near the diagonal point $(b_2, b_3, d) = (0, 0, 0)$, we treat the off-diagonal entries $(d, b_2, b_3)$ as small and use Schur-complement identities for $\mathbb{H}^{-1}$. A direct calculation gives

$$(\mathbb{H}^{-1})_{13} = \frac{-b_2 d + b_3 c}{|\mathbb{H}|}.$$

The FOC $\partial_{b_3}\ell = (\mathbb{H}^{-1})_{13} = 0$ therefore enforces

$$b_3 = \frac{d}{c} b_2,$$

so $b_3$ is slaved to $b_2$ and $d$ in a neighborhood of the diagonal manifold.

We now parameterize the linear response of $b_2$ to $d$ by a coefficient $\alpha$,

$$b_2 = \alpha\, d + O(d^3),$$

and determine $\alpha$ by enforcing the $b_2$–FOC. Substituting $b_3 = (d/c)\, b_2$ and $b_2 = \alpha d$ into $\partial_{b_2}\ell = 0$ and expanding to leading order in $d$ yields

$$\alpha = \begin{cases} \dfrac{ac}{c^2 + ae}, & \text{RNN}, \\ 0, & \text{DNN}. \end{cases}$$

Consequently, near the diagonal manifold,

$$b_2 = \alpha d + O(d^3), \qquad b_3 = \frac{\alpha}{c} d^2 + O(d^4) \quad \text{(RNN)}, \qquad b_2 = b_3 = 0 \quad \text{to these orders (DNN)}.$$

**Quadratic curvature along the interior off-diagonal**  We now substitute

$$b_2 = \alpha d, \qquad b_3 = \frac{d}{c} b_2$$

back into $\ell$ and expand in powers of $d$. The resulting Landau expansion has the form

$$\ell(d) = \ell_0 + \frac{1}{2} C^{(2)}(\alpha)\, d^2 - \frac{1}{4} \frac{d^4}{c^2 e^2} + \cdots, \tag{41}$$

with

$$C^{(2)}(\alpha) = -\frac{1}{ce} + \frac{\alpha}{wac} \quad \Rightarrow \quad \boxed{C^{(2)}_{\text{RNN}} = -\frac{1}{ce} + \frac{1}{w\,(c^2 + ae)}, \qquad C^{(2)}_{\text{DNN}} = -\frac{1}{ce} \ (<0).} \tag{42}$$

The term $-\frac{1}{ce}$ is the entropic penalty from $\ln|\mathbb{H}|$ (Hadamard inequality: at fixed diagonal, the determinant $|\mathbb{H}|$ is maximized when off-diagonals vanish). The quartic coefficient $-1/(4c^2 e^2)$ is architecture-independent and makes $-\ell$ locally stabilizing.

**Critical diagonals for the RNN**  A continuous transition occurs when the quadratic coefficient $C^{(2)}_{\text{RNN}}$ changes sign. The critical point is defined by

$$C^{(2)}_{\text{RNN}} = 0 \quad \Longleftrightarrow \quad w(c^2 + ae) = ce.$$

Combining this with the diagonal relations (39),

$$a^2 = \frac{u}{w} c, \qquad e = \frac{c^2}{a},$$

one obtains the criticality condition for three variables

$$\boxed{a^* = 2\,u, \qquad c^* = 4\,uw, \qquad e^* = 8\,uw^2.} \tag{43}$$

At $(a^*, c^*, e^*)$ the RNN is marginal in the direction of the interior off-diagonal $d$. In contrast, in the DNN one always has $C^{\text{DNN}}_2 < 0$, so the mode remains strictly massive and no such transition occurs.

**Exact critical label strength in terms of $\lambda$**   Plugging (43) into the $e$–FOC (40) gives at criticality

$$\frac{1}{e^*} - \frac{1}{wc^*} + \frac{y_4^{*\,2}}{v\left(e^*\right)^2} = 0.$$

Using $c^* = 4uw$ and $e^* = 8uw^2$, one has $1/(wc^*) = 2/e^*$, so the bracket simplifies to

$$-\frac{1}{e^*} + \frac{y_4^{*\,2}}{v\left(e^*\right)^2} = 0 \quad \Longleftrightarrow \quad y_4^{*\,2} = ve^* = 8\,u\,v\,w^2.$$

Equivalently, in terms of the dimensionless control parameter (37),

$$\lambda^* = \frac{y_4^{*\,2}}{u\,w^2\,v} = 8. \tag{44}$$

Since the label enters only through $y_4^2$, the sign of $y_4$ is irrelevant for the transition. For $\lambda < 8$, one has $C_{\mathrm{RNN}}^{(2)} < 0$ and the unique maximizer is $d^* = 0$; for $\lambda > 8$, one has $C_{\mathrm{RNN}}^{(2)} > 0$ and two symmetric non-zero maximizers $\pm d^* \neq 0$ appear.

**Scaling near criticality without auxiliary variables**   Eliminating the diagonal displacement in favor of $\lambda - 8$ gives a linear expansion

$$C^{(2)}(\lambda) = K_\lambda\,(\lambda - 8) + o(\lambda - 8), \qquad K_\lambda = \frac{1}{1280\,u^2\,w^3}.$$

Maximizing $\ell(d)$ in (41) away from $d = 0$ yields

$$d^{*\,2} = C^{(2)}\,c^2e^2.$$

Evaluating $c^2e^2$ at the critical diagonals (43) gives $c^{*\,2}e^{*\,2} = 1024\,u^4w^6$, hence

$$d^{*\,2} = \frac{4}{5}\,u^2\,w^3\,(\lambda - 8) + o(\lambda - 8) \qquad (\lambda \downarrow 8^+). \tag{45}$$

Equivalently, $d^* = \pm\sqrt{\frac{4}{5}u^2w^3(\lambda - 8)} + o\left(\sqrt{\lambda - 8}\right)$, i.e. the usual mean-field exponent $\beta = \frac{1}{2}$ for $d$.

For the concrete choice $u = v = w = 1$, one has $\lambda = y_4^2$, $\lambda^* = 8$, and

$$d^{*\,2} = \frac{4}{5}\,(\lambda - 8) + o(\lambda - 8).$$

### A.6. Numerical solution of kernels for saddle point equations

We here describe how to obtain the maximum a posteriori kernels that describe converged training for the linear kernels $\mathbb{H}$ (7) and nonlinear kernels $\Phi, \tilde{C}$ (3). For both solvers, we initialize the kernels at the NNGP solution and add a slight off-diagonal for $\mathbb{H}$ and $\Phi$ to break symmetry.

**Linear case**   We use SciPy's Newton-CG optimizer, a second-order method that leverages Hessian-vector products for efficient curvature approximation without forming the full Hessian. Gradients and Hessian-vector products are computed via automatic differentiation in JAX.

**Nonlinear case**   The GD solver iteratively refines the kernels $(\Phi, \tilde{C})$ by computing gradients of the free energy $S :=$ $\ln P(\Phi, \tilde{C})$ through two passes: a "*forward pass*" that evaluates the feature moments at the *equilibrium* kernels $(\mathbb{H}_{\mathrm{eql}}^+ = \langle h^+ h^{+\mathsf{T}} \rangle_{(\Phi, \tilde{C})}, \Phi_{\mathrm{eql}}^+ = \langle \phi(h^+)\phi(h^+)^{\mathsf{T}} \rangle_{(\Phi, \tilde{C})})$ under a tilted Gaussian measure parameterized by the current solver state $(\Phi, \tilde{C})$, and a "*backward pass*" that computes the equilibrium dual kernels $\tilde{C}_{\mathrm{eql}}$ via matrix inversions. The gradients take the symmetric form

$$\nabla_\Phi S = \tilde{C} - \tilde{C}_{\mathrm{eql}},$$
$$\nabla_{\tilde{C}} S = \Phi - \Phi_{\mathrm{eql}}.$$

We perform gradient *descent* in $\Phi$, projecting the gradient to preserve positive definiteness, and gradient *ascent* in $\tilde{C}$. We perform these updates using an extragradient scheme that recomputes gradients at half steps.

A.6.1. COMPUTATIONAL COMPLEXITY

The computation of the gradients works with matrices of shape $\mathbb{R}^{TP \times TP}$. This dictates the fundamental computational cost. In particular, the algorithm requires matrix inversions, which scale as $\mathcal{O}\left((PT)^3\right)$. For the nonlinear case, Monte Carlo sampling of the expectations $\langle \phi(h)\phi(h)^\top \rangle$ is necessary in addition.

## A.7. Overview table of mean-field derivation

In this section, we give a compact overview of the derivation in Section A.3.1 to give a clearly traceable route through the derivation, with definitions given there. In particular, this will reveal why the chosen intensive scaling of the prior variances yields a well-defined set of saddle point equations, explaining why the chosen order parameters concentrate. We will adopt a "greedy" version of Einstein summation convention, where summations should be placed as tightly as possible around repeated indices. We will here not explicitly write out contractions over indices $p = 1 \ldots P$ for clarity. We will also assume $P, T \ll N$ and drop any such subleading terms.

Below we make a step-by-step derivation of the network's prior predictive distribution (14)

$$P(y|\boldsymbol{x}) = \int d\boldsymbol{U}\, d\boldsymbol{W}\, d\boldsymbol{V}\ P(\boldsymbol{U})\, P(\boldsymbol{W})\, P(\boldsymbol{V})$$

$$\times \int d\boldsymbol{h}\, d\xi\ P(\xi^t) \prod_t^T P(y^t \mid \boldsymbol{h}^{t-1}, \boldsymbol{V}, \xi^t) \prod_t^T P(\boldsymbol{h}^t \mid \boldsymbol{h}^{t-1}, \boldsymbol{x}^{t-1}, \boldsymbol{U}, \boldsymbol{W}),$$

where $\boldsymbol{h} \in \mathbb{R}^{T \cdot P \times N}, \xi \in \mathbb{R}^{TP}$.

$$
\begin{aligned}
&P(y|\boldsymbol{x}) \\
&= \int_{\boldsymbol{h}\xi} \int_{\boldsymbol{UWV}} \quad \delta\Big( y^t \quad -(\boldsymbol{V}\boldsymbol{\phi}^{t-1} \quad +\xi^{t-1}) \Big) \\
&\qquad\qquad\qquad \delta\Big( \boldsymbol{h}^t \quad -(\boldsymbol{W}^{(t-1)}\boldsymbol{\phi}^{t-1} \quad +\boldsymbol{U}\boldsymbol{x}^{t-1}) \Big) \\
&\qquad\qquad\qquad \mathcal{N}_{\boldsymbol{U}}(\boldsymbol{U})\mathcal{N}_{\boldsymbol{W}}(\{\boldsymbol{W}^{(t)}\}_t) \quad \mathcal{N}_{\boldsymbol{V}}(\boldsymbol{V}) \quad \mathcal{N}_{\mathsf{K}}(\xi)
\end{aligned}
$$

$$
\begin{aligned}
&\overset{1.}{=} \int_{\boldsymbol{h}\xi\tilde{\boldsymbol{h}}\tilde{f}} \int_{\boldsymbol{UWV}} \quad \exp\big\{ \iota\tilde{f}^t y^t \quad -\iota\tilde{f}^t \boldsymbol{V}\boldsymbol{\phi}^{t-1} \quad -\iota\tilde{f}^t\xi^t \big\} \\
&\qquad\qquad\qquad \exp\big\{ \iota\tilde{\boldsymbol{h}}^t\boldsymbol{h}^t \quad -\iota\tilde{\boldsymbol{h}}^t\boldsymbol{W}^{(t-1)}\boldsymbol{\phi}^{t-1} \quad -\iota\tilde{\boldsymbol{h}}^t\boldsymbol{U}\boldsymbol{x}^{t-1} \big\} \\
&\qquad\qquad\qquad \mathcal{N}_{\boldsymbol{U}}(\boldsymbol{U})\mathcal{N}_{\boldsymbol{W}}(\{\boldsymbol{W}^{(t)}\}_t) \quad \mathcal{N}_{\boldsymbol{V}}(\boldsymbol{V}) \quad \mathcal{N}_{\mathsf{K}}(\xi)
\end{aligned}
$$

$$
\begin{aligned}
&\overset{2.}{=} \quad \int_{\boldsymbol{h}\tilde{\boldsymbol{h}}\tilde{f}} \quad \exp\big\{ \iota\tilde{f}^t y^t \quad -\tfrac{1}{2}\tilde{f}^t(V\phi_i^{t-1}\phi_i^{t'-1})\tilde{f}^{t'} \quad -\tfrac{1}{2}\tilde{f}^t\mathsf{K}\delta^{tt'}\tilde{f}^{t'} \\
&\qquad\qquad\qquad\qquad \iota\tilde{h}_i^t h_i^t \quad -\tilde{h}_i^t(W\delta^{ts}\phi_j^{t-1}\phi_j^{t'-1})\tilde{h}_i^{t'} \quad -\tfrac{1}{2}\tilde{h}_i^t(Ux_j^{t-1}x_j^{t'-1})\tilde{h}_i^{t'} \big\}
\end{aligned}
$$

$$
\begin{aligned}
&\overset{3.}{=} \int_{\Phi\tilde{C}} \exp\big\{ -\tilde{C}^{tt'}\Phi^{tt'} \big\} \int_{\boldsymbol{h}\tilde{\boldsymbol{h}}\tilde{f}} \quad \exp\big\{ \iota\tilde{f}^t y^t \quad -\tfrac{1}{2}\tilde{f}^t(VN\Phi^{t-1,t'-1})\tilde{f}^{t'} \quad -\tfrac{1}{2}\tilde{f}^t\mathsf{K}\delta^{tt'}\tilde{f}^{t'} \\
&\qquad\qquad\qquad\qquad\qquad\qquad \iota\tilde{h}_i^t h_i^t \quad -\tfrac{1}{N}\phi_i^t\tilde{C}^{tt'}\phi_i^{t'} \quad -\tilde{h}_i^t(NW\ _{\tilde{\delta}}{}^{tt'}\Phi^{t-1,t'-1})\tilde{h}_i^{t'} -\tfrac{1}{2}N\tilde{h}_i^t(DU\mathbb{X}^{t-1,t'-1})\tilde{h}_i^{t'} \big\}
\end{aligned}
$$

$$
\begin{aligned}
&\overset{4.}{=} \int_{\Phi\tilde{C}} \exp\big\{ -\tilde{C}^{tt'}\Phi^{tt'} \big\} \int_{\boldsymbol{h}\tilde{\boldsymbol{h}}\tilde{f}} \quad \exp\big\{ \iota N\tilde{f}^t y^t \quad -\tfrac{1}{2}N^2\tilde{f}^t(VN\Phi^{t-1,t'-1})\tilde{f}^{t'} \quad -\tfrac{1}{2}N^2\tilde{f}^t\mathsf{K}\delta^{tt'}\tilde{f}^{t'} \\
&\qquad\qquad\qquad\qquad\qquad\qquad \iota\tilde{h}_i^t h_i^t \quad -\tfrac{1}{N}\phi_i^t\tilde{C}^{tt'}\phi_i^{t'} \quad -\tilde{h}_i^t(NW\ _{\tilde{\delta}}{}^{tt'}\Phi^{t-1,t'-1})\tilde{h}_i^{t'} -\tfrac{1}{2}N\tilde{h}_i^t(DU\mathbb{X}^{t-1,t'-1})\tilde{h}_i^{t'} \big\}
\end{aligned}
$$

$$
\begin{aligned}
&\overset{5.}{=} \int_{\Phi\tilde{C}} \exp\big\{ -\tilde{C}^{tt'}\Phi^{tt'} \big\} \int_{\tilde{f}} \quad \exp\big\{ \iota N\tilde{f}^t y^t \quad -\tfrac{1}{2}N^2\tilde{f}^t(VN\Phi^{t-1,t'-1})\tilde{f}^{t'} \quad -\tfrac{1}{2}N^2\tilde{f}^t\mathsf{K}\delta^{tt'}\tilde{f}^{t'} \big\} \\
&\qquad\qquad\qquad\qquad\qquad \times \int_{\boldsymbol{h}\tilde{\boldsymbol{h}}} \prod_i \exp\big\{ \iota\tilde{h}_i^t h_i^t \quad \tfrac{1}{N}\phi_i^t\tilde{C}^{tt'}\phi_i^{t'} \quad -\tilde{h}_{t,i}(NW\ _{\tilde{\delta}}{}^{tt'}\Phi^{t-1,t'-1})\tilde{h}_i^{t'} -\tfrac{1}{2}N\tilde{h}_i^t(DU\mathbb{X}^{t-1,t'-1})\tilde{h}_i^{t'} \big\}
\end{aligned}
$$

$$
\begin{aligned}
&\overset{6.}{=} \int_{\Phi\tilde{C}} \exp\big\{ -\tilde{C}^{tt'}\Phi^{tt'} \big\} \int_{\tilde{f}} \quad \exp\big\{ \iota N\tilde{f}^t y^t \quad -\tfrac{1}{2}N^2\tilde{f}^t(VN\Phi^{t-1,t'-1})\tilde{f}^{t'} \quad -\tfrac{1}{2}N^2\tilde{f}^t\mathsf{K}\delta^{tt'}\tilde{f}^{t'} \big\} \\
&\qquad\qquad\qquad\qquad\qquad \times \int_{\boldsymbol{h}\tilde{\boldsymbol{h}}} \prod_i \exp\big\{ \iota\tilde{h}^t h^t \quad \phi^t\tilde{C}^{tt'}\phi^{t'} \quad -\tilde{h}^t(NW\ _{\tilde{\delta}}{}^{tt'}\Phi^{t-1,t'-1})\tilde{h}^{t'} -\tfrac{1}{2}\tilde{h}^t(DU\mathbb{X}^{t-1,t'-1})\tilde{h}^{t'} \big\}
\end{aligned}
$$

$$
\begin{aligned}
&\overset{7.}{=} \int_{\Phi\tilde{C}} \exp\big\{ -N\tilde{C}^{tt'}\Phi^{tt'} \big\} \int_{\tilde{f}} \quad \exp\big\{ \iota N\tilde{f}^t y^t \quad -\tfrac{1}{2}N\tilde{f}^t(v\Phi^{t-1,t'-1})\tilde{f}^{t'} \quad -\tfrac{1}{2}N\tilde{f}^t\kappa\delta^{tt'}\tilde{f}^{t'} \big\} \\
&\qquad\qquad\qquad\qquad\quad + N\ \ln\big[\int_{h\tilde{h}} \exp\big\{ \iota\tilde{h}^t h^t \quad \phi^t\tilde{C}^{tt'}\phi^{t'} \quad -\tilde{h}^t(w\ _{\tilde{\delta}}{}^{tt'}\Phi^{t-1,t'-1})\tilde{h}^{t'} \quad -\tfrac{1}{2}\tilde{h}_i^t(u\mathbb{X}^{t-1,t'-1})\tilde{h}_i^{t'} \big\}\big]\big\}
\end{aligned}
$$

$$
\begin{aligned}
&\overset{8.}{=} \int_{\Phi\tilde{C}} \exp\big\{ -N\tilde{C}^{tt'}\Phi^{tt'} \big\} \quad \exp\big\{ \quad -\tfrac{1}{2}y^t N\big((v\Phi^- + \kappa)^{-1}\big)^{tt'} y^{t'} \quad -\tfrac{1}{2}\ln\big|\tfrac{1}{N}(v\Phi^- + \kappa)\big| \\
&\qquad\qquad\qquad\qquad\quad + N\ \ln\big[\int_{h\tilde{h}} \exp\big\{ \quad \phi^t\tilde{C}^{tt'}\phi^{t'} \quad -\tfrac{1}{2}h^t\big((w[\![\Phi^-]\!] + u\mathbb{X}^-)^{-1}\big)^{tt'} h^{t'} -\tfrac{1}{2}\ln\big|w[\![\Phi^-]\!] + u\mathbb{X}^-\big| \quad \big\}\big]
\end{aligned}
$$

$$
\begin{aligned}
&\overset{9.}{=} \int_{\Phi\tilde{C}} \exp\big\{ N\big[-\tilde{C}^{tt'}\Phi^{tt'} \quad -\tfrac{1}{2}y^t\big((v\Phi^- + \kappa)^{-1}\big)^{tt'} y^{t'} \quad -\tfrac{1}{2}\tfrac{1}{N}\ln\big|\tfrac{1}{N}(v\Phi^- + \kappa)\big| \\
&\qquad\qquad\qquad\qquad\quad - \quad \ln\ Z(\tilde{C};\Phi) \quad \big]\big\}
\end{aligned}
$$

Herein, we made the following manipulations:

1. Rewrite each Dirac delta enforcing the deterministic network equations in its Fourier representation, $\delta(\circ) = \int d\tilde{z}\,\exp\{\iota\tilde{z}(\circ)\}$, introducing conjugate fields $\tilde{f}^t, \tilde{h}^t$ for the outputs $y^t$ and hidden pre-activations $h^t$. This converts the constraints into linear couplings in the exponent, such that all random variables only appear in the equation through quadratic forms.

2. Integrate out the Gaussian weight priors $\{\boldsymbol{W}^{(t)}\}_t, \boldsymbol{U}, \boldsymbol{V}$ and the output noise $\xi$. We introduce the variable $\bar{\delta}^{tt'}$

$$\bar{\delta}^{tt'} := \begin{cases} \delta^{tt'}, & \text{DNN} \\ 1, & \text{RNN} \end{cases}$$

which allows us to write down the covariance of the priors in the short form $\langle \boldsymbol{W}_{ij}^{(t)} \boldsymbol{W}_{ij}^{(t')} \rangle = W\,\bar{\delta}^{tt'}$ (and analogously for $U, V, \kappa$). We denote the corresponding masking operator on time indices by

$$(\llbracket A \rrbracket)^{tt'} := \bar{\delta}^{tt'}\, A^{tt'},$$

which acts as a diagonal projector in $t, t'$ for DNNs and as the identity for RNNs.

3. Introduce the empirical kernel $\Phi^{tt'} := \frac{1}{N}\sum_{i=1}^N \phi_i^t \phi_i^{t'}$ via a Dirac delta, $\delta\left(\Phi - \frac{1}{N}\sum_i \phi_i \phi_i\right) = \int d\tilde{C}\,\exp\left\{\tilde{C}^{tt'}\left(\Phi^{tt'} - \frac{1}{N}\sum_i \phi_i^t \phi_i^{t'}\right)\right\}$, in order to decouple different neurons. The $\int d\tilde{C}$ integral is taken along a contour parallel to the imaginary axis passing through the saddle point; we omit an explicit imaginary unit, anticipating that the saddle lies on the real axis. As long as there are no singularities between the original and shifted contours, such deformations are justified (Bromwich contour).

4. Rescale the output conjugate field according to $\tilde{f} \mapsto N\tilde{f}$ so that its quadratic form scales as $\mathcal{O}(N)$, on the same footing as the contribution from the hidden conjugate fields. This is an algebraic change of variables that prepares the expression for a uniform large-$N$ analysis.

5. Exploit the fact that now, conditional on the collective fields $(\Phi, \tilde{C})$, different neurons are independent and identically distributed. This allows us to factor the $N$-dimensional integral over $\{h_i, \tilde{h}_i\}_{i=1}^N$ into a product of identical single-site integrals.

6. Rescale the conjugate kernel to make the overall $N$-dependence explicit, $\tilde{C} \mapsto N\tilde{C}$. After this redefinition the kernel term in the exponent takes the canonical form $-\frac{N}{2}\tilde{C}^{tt'}\Phi^{tt'}$, so that the full effective action is proportional to $N$.

7. Insert the mean-field ('intensive') scalings for the prior variances, e.g. $V = v/N^2$, $W = w/N$, $U = w/N$ (and similarly for the noise level), so that the pre-activation covariances and outputs remain of order one as $N \to \infty$. This yields quadratic forms involving $v, w, u$ and the kernels $\Phi$ rather than the original extensive parameters.

8. Perform the remaining Gaussian integrals over $\tilde{f}$ and over the hidden fields (after integrating out their conjugate fields inside the single-site factor). This produces the quadratic form in the outputs $y$, with covariance $v\Phi^- + \kappa$, as well as the corresponding log-determinant normalization, and an analogous Gaussian contribution for the hidden fields involving the matrix $w\llbracket \Phi^- \rrbracket + u\mathbb{X}^-$.

9. Recognize that the contribution of a single neuron is captured by a partition function $Z(\tilde{C}; \Phi)$ which only depends on the collective fields through $(\Phi, \tilde{C})$. Because all neurons are identically distributed, the full factor from the hidden layer is $Z(\tilde{C}; \Phi)^N$, which yields a term $-N\ln Z(\tilde{C}; \Phi)$ in the effective action.

10. Apply a saddle-point approximation (Laplace's method) to the remaining functional integral over $(\Phi, \tilde{C})$ in the limit of large width $N$. The exponent is of the form $N\,S(\Phi, \tilde{C})$, so the integral is dominated by the stationary points of $S$; this is equivalent to invoking a large-deviation principle for the empirical kernel.

A key object that arises from this calculation is the single-site cumulant generating function

$$W(\tilde{C}; \Phi) = \ln Z(\tilde{C}; \Phi) = \ln \int_{h\tilde{h}} \exp\left\{-\phi^t \tilde{C}^{tt'} \phi^{t'} - \tfrac{1}{2}h^t\left(\left(w\llbracket \Phi^- \rrbracket + u\mathbb{X}^-\right)^{-1}\right)^{tt'} h^{t'} - \tfrac{1}{2}\ln\left|w\llbracket \Phi^- \rrbracket + u\mathbb{X}^-\right|\right\}. \quad (46)$$

whose derivatives with respect to $\tilde{C}$ yield the cumulants $\phi\phi^\mathsf{T}$.

## A.8. Additional figures

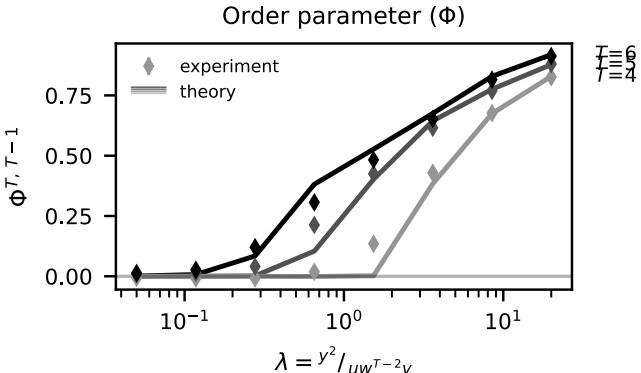

**Figure A.6. Phase transition of kernel temporal coherence order parameter for $\phi(h) = \mathrm{erf}(h)$.** Same as Fig. 4 of the main text but using $\phi = \mathrm{erf}$ instead of a linear activation. Lines are the kernel mean-field theory, markers are weight-space simulations.

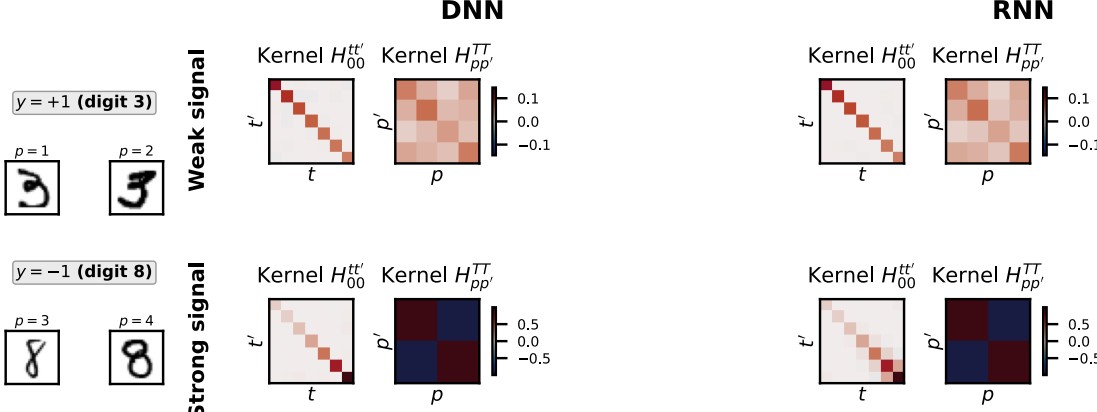

**Figure A.7. RNN vs DNN comparison on MNIST.** Same as Fig. 3 of the main text but with MNIST digits as input instead of orthogonal data. The qualitative kernel structure is preserved.

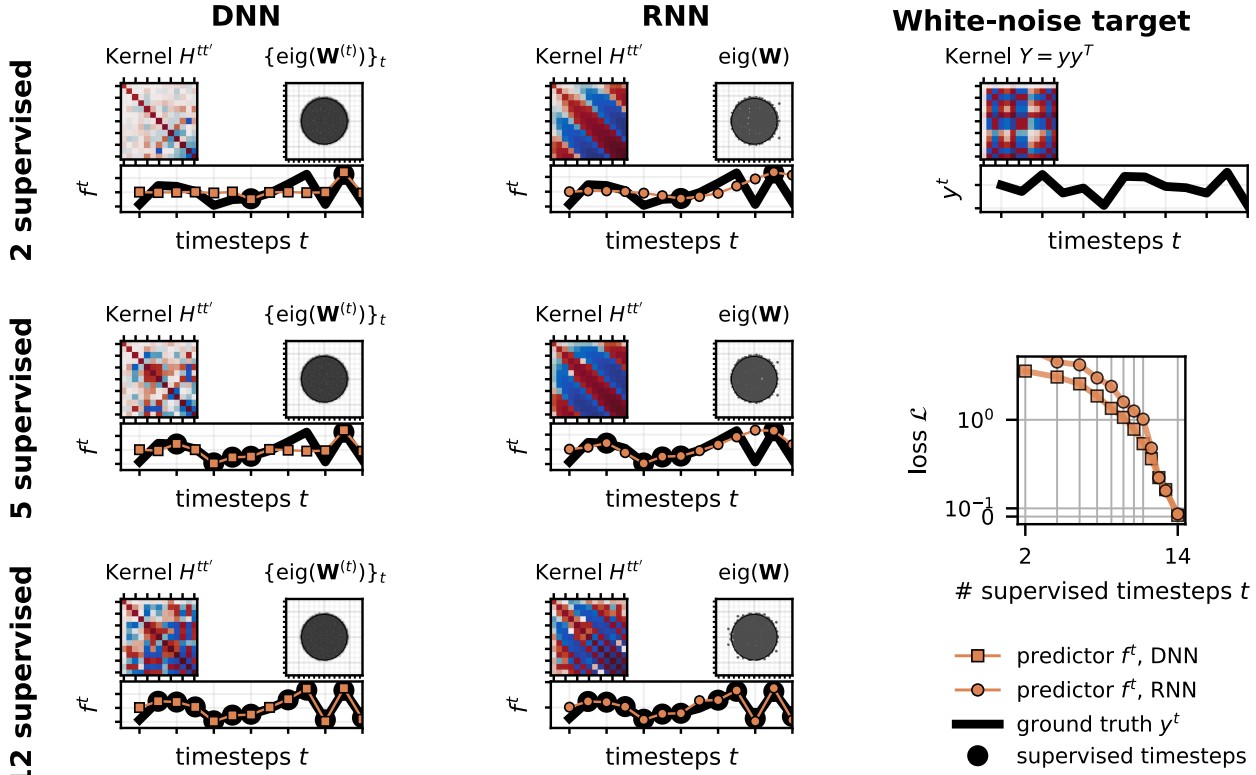

**Figure A.8. White-noise (zero-correlation) target.** Same as Fig. 5 of the main text but with a temporally uncorrelated (white-noise) target instead of the teacher-RNN sequence. Without temporal structure to exploit, the RNN latches onto spurious correlations between supervised timesteps, whereas the DNN's uncorrelated prior aligns with the task and through this effective regularization yields lower loss.

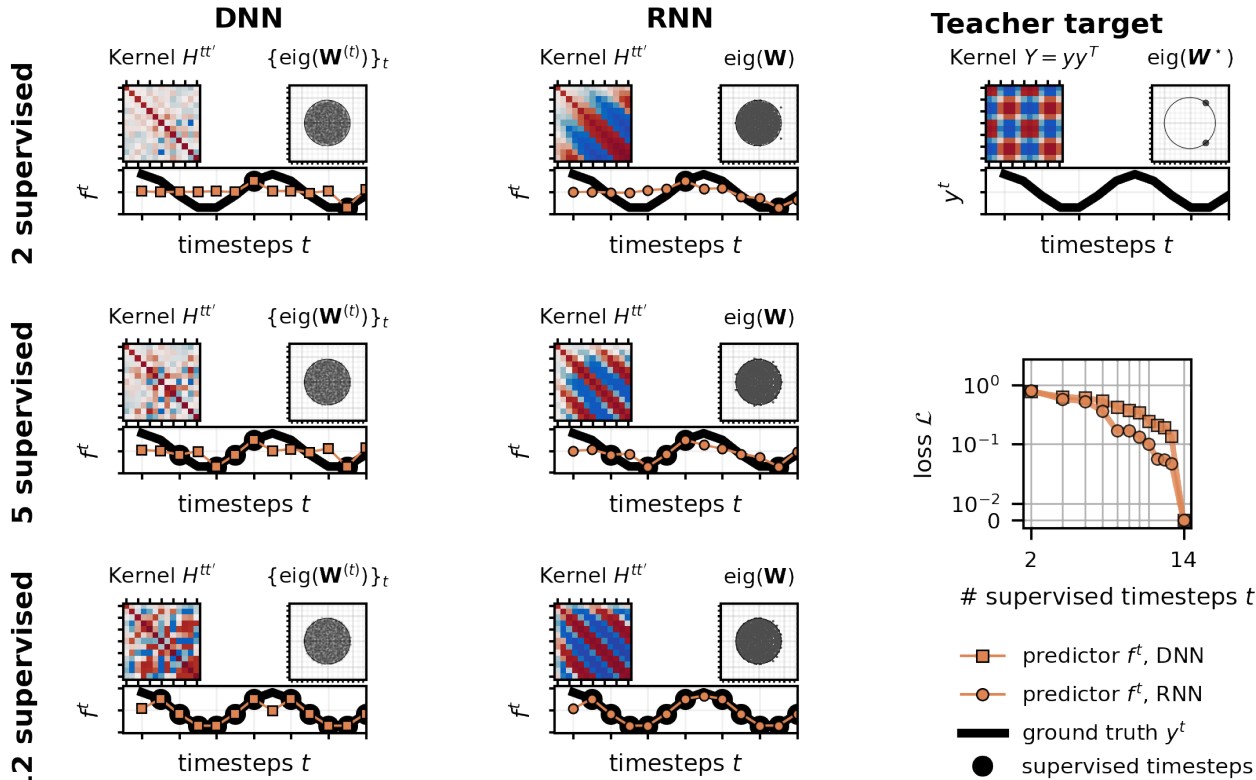

**Figure A.9. RNNs are efficient sequence learners even when parameter-count matched.** Same as Fig. 5 of the main text but with the RNN hidden size increased to $N = 885$ so that both architectures have the same parameter count ($\approx 786$k). The RNN continues to generalize better at sparse supervision.

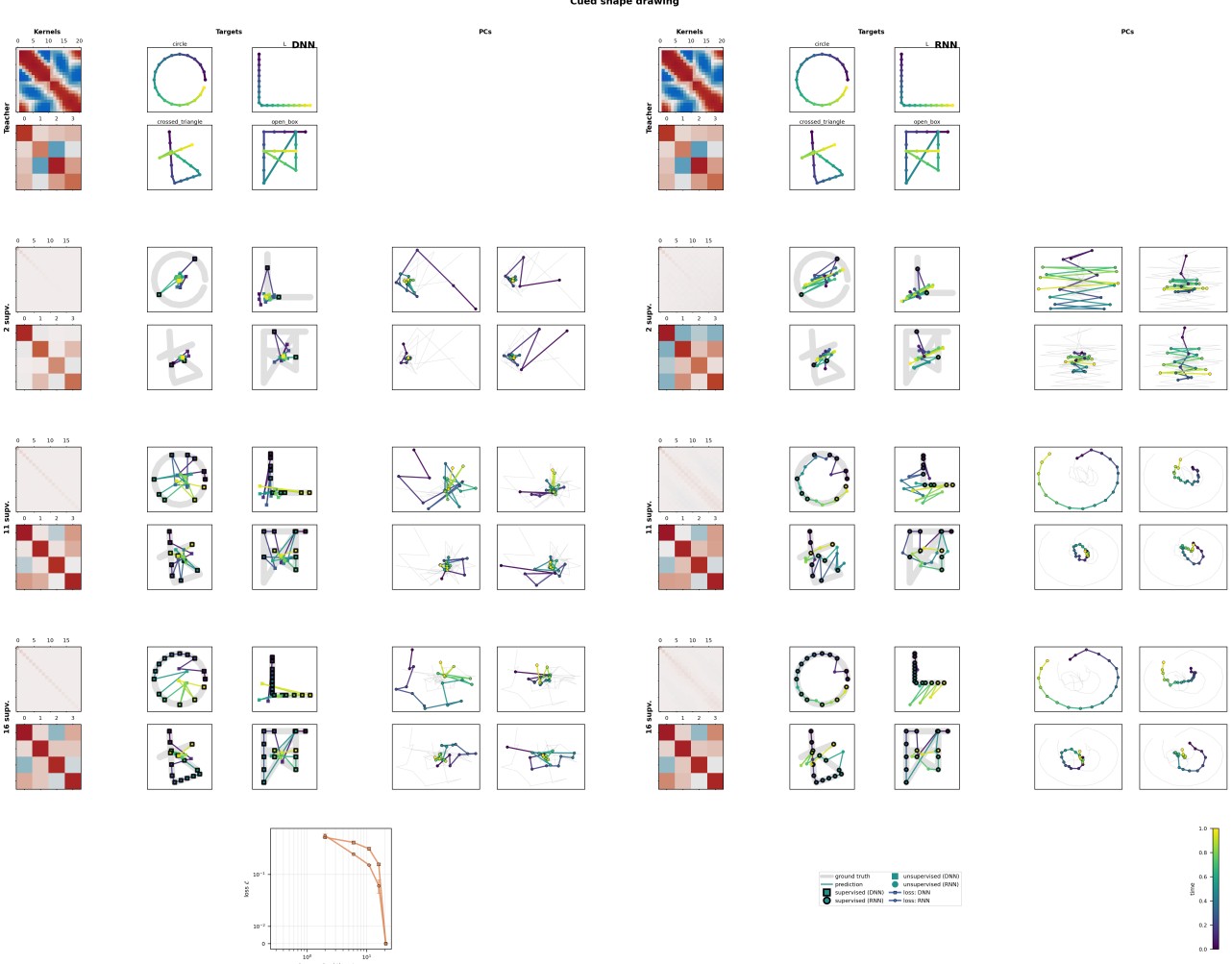

**Figure A.10. RNNs are efficient sequence learners: cued shape-drawing task.** A network receives a one-hot shape cue at $t = 0$ and must produce the 2D drawing trajectory. Four shapes are shown: circle, L, crossed triangle, open box. Left/right halves compare DNN and RNN at increasing supervision density ($n_{\mathrm{supv}} \in \{2, 11, 16\}$ out of 20 timesteps). Each row shows learned kernels, predicted vs. ground-truth trajectories (blue-to-yellow colormap), and top-2 PCs of hidden activity. Bottom: generalization loss vs. number of supervised steps. The RNN generalizes better at intermediate supervision. $N = 256$, $\phi = \mathrm{erf}$, $\kappa = 0$.

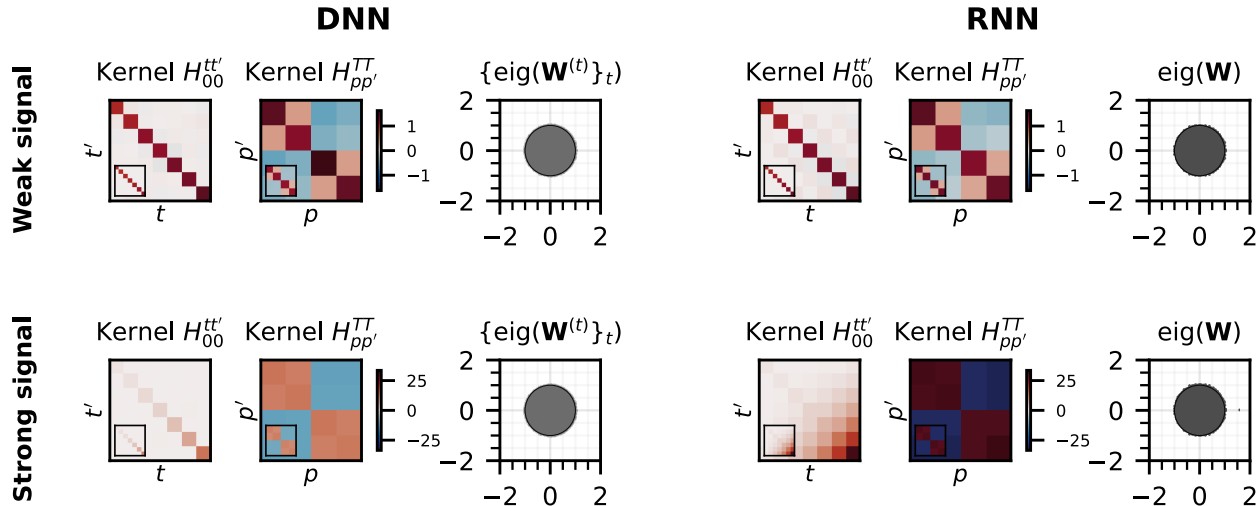

**Figure A.11. RNN vs DNN comparison at zero temperature.** Same as Fig. 3 of the main text but trained with zero-temperature SGLD ($\kappa = 0$). The phase transition towards temporal coherence in the RNN, and its absence in the DNN, persists in this deterministic, noiseless limit.

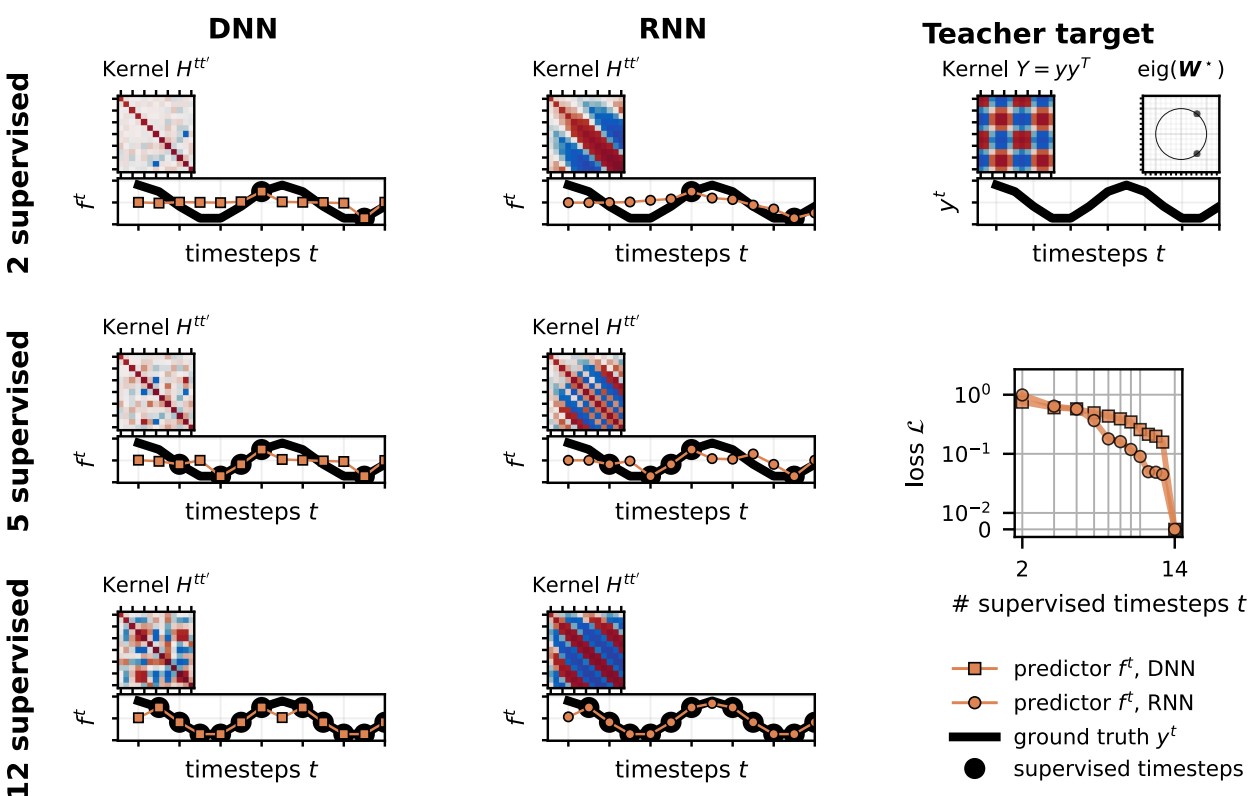

**Figure A.12. RNNs are efficient sequence learners at zero temperature.** Same as Fig. 5 of the main text but with $\phi(h) = \mathrm{erf}(h)$ and zero-temperature SGLD ($\kappa = 0$). The RNN's generalization advantage at sparse supervision persists in this deterministic limit.

