# OpenReview forum: "A unified theory of feature learning in RNNs and DNNs"
_ICML.cc/2026/Conference — ICML 2026 regular_

### Official Review · Reviewer_19Up · 2026-03-03

**Soundness:** 3
**Presentation:** 3
**Significance:** 3
**Originality:** 3
**Overall Recommendation:** 4
**Confidence:** 1

**Summary:**

This paper establishes a mean-field theoretical framework for DNNs and RNNs, using representative kernels at the Maximal update parameterization scale to describe the post-trained state of the network. Based on this, the paper demonstrates that weight sharing in RNNs is an inductive bias in sequence tasks, showing the critical importance of the architecture itself for deep learning tasks.

**Compliance With Llm Reviewing Policy:**

Affirmed.

**Final Justification:**

My concerns have been adequately addressed.

**Key Questions For Authors:**

1. Could the authors provide results where the RNN and DNN have matched total parameter budgets?
2. Given the sine wave's inherent cyclic symmetry, how does the weight-sharing bias behave on non-stationary?
3. How would this framework explain the behavior of Transformers, which lack weight sharing but utilize positional encodings?

**Limitations:**

yes

**Strengths And Weaknesses:**

Strength：
- The training of the network is described as sequence patterns and Bayesian inference, and the commonalities of weight sharing and temporal correlation masks are explained.
- It incorporates RNN and DNN into a unified mathematical framework, provides detailed derivation processes, and conducts relatively thorough experiments.
- From the perspective of feature space alignment and inductive bias, it provides a new understanding of weight sharing, rather than just focusing on reducing parameters.

Weakness
- The paper uses a sine wave regression task, a simple recurrent task that is highly suitable for RNNs and may amplify their advantages.
- According to the architecture described in the paper, the number of parameters in an RNN is much smaller than that of a DNN, so its poor performance on a simple recurrent task is to be expected, and it may exaggerate some findings.
- Modern deep learning typically uses mini-batch processing and less frequently LSGD, reflecting a gap between mathematical models and reality.

---

> ### Author Rebuttal · Authors · 2026-03-31
>
> We thank the reviewer for their review and hope that our replies below address the raised concerns, where we believe one is based on a misunderstanding.
>
> ### **Sine wave task and non-stationarity**
>
> The sine wave task was a deliberate choice to highlight, in a minimal setting with an interpretable teacher signal, the beneficial inductive bias of an RNN, despite its lesser expressivity compared to the DNN. The insights derived from the theory  generalize beyond this specific task: Eqs. (9) and (10) are valid for arbitrary target functions $y^t$, including non-stationary ones, and thus for arbitrary target correlations $\\mathbb{Y}^{tt′}$. Any form of temporal coherence between supervision points, as encoded in $Y$, will be picked up more efficiently by the RNN than the DNN, which can be directly seen from comparing these two equations. To demonstrate the generality of this key mechanism, we have recreated our main findings for real-world, non-stationary tasks \[1,2\], as well as more complicated architectures having the same weight sharing paradigm \[3, cf. reply to pbAS, “**Empirical validation”**\]
>
> ### **Expressivity vs inductive bias**
>
> > According to the architecture described in the paper, the number of parameters in an RNN is much smaller than that of a DNN, so its poor performance on a simple recurrent task is to be expected, and it may exaggerate some findings.
>
> There may be a misunderstanding here: the RNN, despite having fewer parameters (weights being shared across layers) *outperforms* the DNN on a time-series task (cf. Fig 5b, compare square \= DNN to round \= RNN). This may indeed at first seem very surprising, but our theory fully explains this: it is the inductive bias of the RNN, its tendency to favor solutions which have temporal coherence across time, which leads to better interpolation than the DNN, *despite* the latter being more expressive. Quantitatively, this result can be understood from comparing Eqs. (9) and (10). The revised manuscript improves the caption and legend of Fig. 5b and the description of this central result to clarify this point.
>
> > Modern deep learning typically uses mini-batch processing and less frequently LSGD, reflecting a gap between mathematical models and reality.
>
> We kindly refer to the discussion in the response to reviewer **c2LB** and **G1f5**, who raised similar concerns.
>
> > Could the authors provide results where the RNN and DNN have matched total parameter budgets?
>
> We agree that it is good to control for parameter-dependent effects, and have created an additional figure ([Fig. I](https://anonymous.4open.science/r/rebuttal-16a25f/RNNs_are_efficient_sequence_learners_param_matched.png)) comparing RNN and DNN with matched numbers of parameters.
>
> ### **Relation to temporal correlations in transformers**
>
> This is an intriguing question. Transformers have weight sharing via $W\_K$ and $W\_Q$ and can be mapped to RNNs \[Katharopoulos et. al, 2020\], but their complexity makes it unclear whether gradient descent favors temporal correlations as in RNNs. Our preliminary experiments suggest transformers behave closer to DNNs, unless a simpler linear version is used (see Fig. C in response to reviewer **c2LB**). We consider inconclusive evidence given the architecture's complexity, so that theory would be useful.
>
> Conceptually, a similar theoretical approach could be pursued for transformers. The crucial property we expect to carry through is the factorization of preactivations across neurons in the large-width limit (remark after Eq. (25)), expected from kernel limit results (e.g. Hron et al. (2020)). Intuitively, this factorization arises from the CLT, independent priors over $W\_{ij}$, and extensive summations such as $V\_i^t \= \\sum\_j W\_{ij} h\_j^t$. Positional encoding then induces correlations between $h^{t}$ and $h^{t’}$ as a function of $t,t’$, analogous to input correlations in our RNN theory (entering the input kernel $\\mathbb{X}$, Eq. (3)) than to direct weight-sharing effects. We thank the reviewer for this intriguing question\!

---

> > ### Author Rebuttal · Reviewer_19Up · 2026-04-02
> >
> > Thanks for your detailed response. I have no further questions.

---

> > > ### Author Response · Authors · 2026-04-03
> > >
> > > We are glad that you found our reply helpful. If it addresses your initial concerns, we would appreciate if you reconsidered your score. In any case, we are very happy to help with further clarifications.

---

### Official Review · Reviewer_pbAS · 2026-03-11

**Soundness:** 3
**Presentation:** 2
**Significance:** 3
**Originality:** 3
**Overall Recommendation:** 4
**Confidence:** 3

**Summary:**

This paper provides a mean-field analysis for feature learning in RNNs and counterparts without weight sharing in the Bayesian Inference posterior with a MuP prior.
Although weight sharing strictly hurts expressivity, weight sharing gives an additional structural prior in RNNs.
The authors have identified a phase transition for a class of "endpoint supervision" problems where RNN induces a "temporal correlation" behavior (hidden pre-activations are positively aligned) when learning signal is sufficiently large, whereas such correlation does not exist without weight sharing. For sequential modelling tasks, the authors showed that RNN's kernel equations naturally generate propagators interpolating the learning signal, which induces inductive bias that aids generalization.

**Compliance With Llm Reviewing Policy:**

Affirmed.

**Final Justification:**

As in the earlier review, I think this paper studies a theoretically motivated framework and gave relatively clear results. Therefore I recommend (weak) acceptance.

The reason for not giving a higher score is due to (IMHO) poor presentation in the initial submission (while the authors have committed to improvements, I cannot verify it). I also remain cautious regarding the actual impact of experiments, but I think the theory alone suffices for an acceptable paper.

**Key Questions For Authors:**

1. Can there be an empirical validation on real datasets for the results concerning RNNs, as weight sharing is indeed empirically verified to be beneficial in may cases? Although I'm not familiar with how successful simulations in LSGD / Bayesian Inference can be carried out, but surely demonstrating the temporal alignments and RNN inductive bias in real data will be interesting.

2. Given a dataset, can your theory predict whether weight sharing will be beneficial (in any relevant sense e.g. Bayesian evidence, generalization error) without actually training the model? A concrete message like "if this moment condition of our dataset is met, then we can predict whether training with RNN will be better or not" will be interesting.

**Limitations:**

yes

**Strengths And Weaknesses:**

Strengths:
1. The question studied (what is the effect of weight sharing in RNN) is important both theoretically and empirically. The message of the paper is clear ("temporal correlation happens when signal strength is large") and well-explained by the theory.
2. The theory is particularly elegent by extracting a clean effect in weight sharing through masking in the kernel space.

Weaknesses:
1. My main objection is concerning the writing. Just to list a non-conclusive few points: (a) the abbrieviation and definition of "DNN" is non-standard and remains confusing until eqn(1, 6). (b) The math feels like it was not properly explained in the main text. Notations in Sec 3.1 (especially eqn(3)) is neither really defined nor explained until in the Appendix. "In this theory, RNNs and DNNs only differ by a masking operation in time $\Phi^{-} = ...$" why is the kernel without weight sharing simply the diagonalization? (c) Several definitions seem important but was not highlighted as definitions. For instance, the notions of "temporally-coherent" and "learning signal strength" are important to the main results but are not mathematically defined or sufficiently explained (what is the meaning of $\lambda = y^2/vw^{T-2}u$, and why is $y$ a scalar here?).

2. While empirical results are present, the plots mostly serve as a demonstration of concept rather than empirical validations, with minimal synthetic examples and modest experiments on toy problems.

Overall, I think the paper's theoretical results are potentially strong and I am willing to raise my score if the authors can meaningfully improve the writing and make both the setup and results understandable in the main text without heavy reference to the appendix for definitions.

---

> ### Author Rebuttal · Authors · 2026-03-31
>
> We are glad that the reviewer found the question studied in our work relevant and the theoretical contributions insightful. We appreciate the constructive comments, and address them below:
>
> ### **Improved presentation of definitions and mathematical derivations**
>
> a) the revised version now introduces all abbreviations at their first occurrence, omitting/delaying the definition of DNN was an oversight.
>
> b) We have re-organize the theory section to include the following conceptual steps for clarity: 1.) The log partition function over the weights describes the posterior over parameters; 2.) computation of integrals over weights 3.) identification of  the order parameter $\\Phi$ 4.) demonstration that this order parameter concentrates, i.e. $P(\\Phi) \\sim \\exp(- N S(\\Phi) )$, where the factor $N$ (hidden layer dimension) comes from the mean-field theory factorizing over neurons, the cause of the concentration 5.) determine $\\Phi$ as saddle point. Eq. (4) describes the distribution of preactivations, which factorize conditioned on this concentrating order parameter (fixing it to a given value).
>
> >  why is the kernel without weight sharing simply the diagonalization?
>
> Indeed, this is by no means obvious to see, being a central outcome of our theory. However, it is possible to provide some intuition: in the DNN, weights are independent across layers. In contrast, in the RNN, weights across time are perfectly correlated (as they are the identical). On the representational level described by the theory, the DNN’s independence is translated into a masked activation kernel $\\text{diag}(\\mathbb{H}^{tt’})=\\text{diag}(\\langle h^t h^{t’}\\rangle )\\propto \\delta^{tt’}$. In essence, independent weights induce independent activations, expressed by the masking.
>
> c) “Temporally coherent” refers to a network state where the preactivations are correlated across different time points (the $T\\times T$ kernel has non-zero values off-diagonal); “temporally incoherent” refers to a network state where cross-time correlations are zero (the kernel is diagonal as in the DNN). The control parameter $\\lambda=||y||^2/vw^{T-2}u$ is a learning signal strength, which takes the form of a signal-to-noise ratio. The signal here is the norm of the labels being learned ($||y||^2=y^2$ for $y \\in \\mathbb{R}$), and noise is the product of the variance of the initialization weights $U,W,V$. We have clarified this point in the manuscript, and made sure that core concepts are well defined and properly explained.
>
> ### **Empirical validation on more realistic tasks**
>
> While we pursued transparency in our choice of tasks, we agree it is important to see whether the results carry over to more realistic settings.
>
> For endpoint-supervised tasks, we now considered MNIST binary classification to distinguish the similar digits 8 and 3 and show that indeed the transition between temporally incoherent and temporally coherent regimes happens as the target scale increases ([Fig. E](https://anonymous.4open.science/r/rebuttal-16a25f/RNNs_vs_DNNs_endpoint_sv_MNIST.png)).
>
> We also provide numerical evidence for the advantageous inductive bias of RNN on two real-world sequence processing tasks. The first one ([Fig. F](https://anonymous.4open.science/r/rebuttal-16a25f/RNNs_are_efficient_sequence_learners_drawings.png)) was originally conceived to study compositional generalization in the primate brain \[Tian et. al, 2026\]. In the second one ([Fig. G](https://anonymous.4open.science/r/rebuttal-16a25f/cyclic_EMG_data.png)), we train both RNNs and DNNs to generate muscle activity recorded from the primate arms \[Saxena et. al, 2024\]. These sequential tasks are in line with the initial, synthetic task: RNNs are more sample efficient (see also our reply to **19up** and Fig. C in the response to reviewer c2BL for how our results generalize to non-RNN architectures with weight sharing).
>
> ### **Theory can predict an architecture’s generalization through task-model alignment**
>
> While predicting the outcome of neural network training is hard, our theory provides some steps into this direction: By reformulating networks as probabilistic models, their inductive bias is revealed: the learned kernel’s correlation will determine generalization behavior. Only if this matches the correlation of the ground truth task being learned, this bias will be beneficial. To illustrate that ground truth correlation length indeed is predictive of generalization behavior, we now trained RNNs and DNNs on tasks of different correlation length ([Fig. H](https://anonymous.4open.science/r/rebuttal-16a25f/generalization_gap_vs_correlation_length.png)). Indeed, the RNN’s advantage increases with correlation length. In contrast, short correlation length puts the DNN at an advantage, since its kernel better describes the approximately uncorrelated data, while the RNN overfits to spurious correlations. This underlines the importance of task-model alignment for learning.

---

> > ### Author Rebuttal · Reviewer_pbAS · 2026-03-31
> >
> > Thank you for addressing my points. I'm willing to raise my score to 4.

---

### Official Review · Reviewer_G1f5 · 2026-03-11

**Soundness:** 3
**Presentation:** 3
**Significance:** 3
**Originality:** 3
**Overall Recommendation:** 5
**Confidence:** 4

**Summary:**

This paper provides a unified mean field theory of trained DNNs and RNNs on supervised learning tasks. This theory is based on equilibrium configurations of large width networks after training with Langevin gradient descent. The authors are able to make a precise comparison of the final representation kernels for models with and without weight sharing. For endpoint supervised tasks (common for deep networks), the RNNs learned representation is the same as DNN for weak supervision but can eventually differ from the DNN for strong enough supervision. The authors argue that RNNs are more sample efficient in the temporal full temporal supervision setting where targets are provided at each timestep.

**Compliance With Llm Reviewing Policy:**

Affirmed.

**Final Justification:**

This paper studies the mean field behavior of trained RNNs and compares their inductive bias to MLPs where there is no weight sharing across layers. I think this is a fundamental question in deep learning theory and I support acceptance. The rebuttal addressed my concerns about relation to SSMs, some of the simulations and missing theory curves, and citations of prior relevant work.

**Key Questions For Authors:**

1. In Figure 3 and 4 what is the nonlinearity? Are these linear RNNs? Stating which figures are linear vs nonlinear could be useful.
2. Do the authors have any numerical solutions to the nonlinear versions of their equations? My impression was that
3. Are there theory curves on Figure A.6? They are not currently visible.

**Limitations:**

The authors could mention the challenges of solving the nonlinear theory for large $T$ and large $P$ (such as the computational complexity) and also discuss that the theory relies on perfect equilibration of LSGD.

**Strengths And Weaknesses:**

***Strengths***

**Concise Unified Framework**

The authors can neatly handle the impact of weight sharing on the learned solution through the masking operator $\left[ \right]$ in time. This allows them to neatly handle both the RNN and DNN cases within a single theory, similar to the prior work of Segadlo et al 2023.

**Exposing Different Inductive Biases**

The strong supervision regime reveals that RNNs can learn qualitatively different solutions, even on endpoint only supervised tasks. Figure 3 demonstrates the emergence of an outlier eigenvalue in the recurrent weights and different structure in the offdiagonal (in time) components of the kernels int he trained RNN, while these are absent in the DNN. Figure 5 highlights a related discrepancy between the DNN and RNNs learned solutions.

**Phase Transition Picture**

The authors argue that there is a critical signal-to-prior ratio that sets the emergence of off diagonal components of the kernel. They argue that this is reminiscent of the BBP phase transition for spiked random matrices. This is a very interesting observation.

***Weaknesses***

**Fixed Point of Langevin Training instead of Real SGD**

The theory hinges on the equilibrium configurations of Langevin training. This requires noise with isotropic covariance, which does not coincide with minibatch SGD which is used in practice.

**No Gating or State Space Model Connections**

The current theory focuses on vanilla RNNs, rather than gated RNNs which enable more interesting computations over long sequences and are commonly used in state of the art sequence / transformer-hybrid models.  This could be out of scope potentially, but I think this connection could be interesting to explore using theory like the one in the paper.

**Limited Solutions of the Theory**
Many of the solutions focus on linear networks learning simple tasks.

**Some Ungenerous Citations**

The authors cite Bordelon et al 2024 Cosyne abstract saying that "Bordelon et al. (2024) studied RNNs in continuous time with a leak term."  This work also studied Bayesian RNN training at large width and derived a mean field theory for the adapted correlation function under feature learning as a function of a tunable parameter as well as a perturbative solution and nonperturbative exact analytical solution in linear case. I think this citation could be a bit more generous as many of the theoretical results of the present work.

---

> ### Author Rebuttal · Authors · 2026-03-31
>
> We are delighted by the reviewer’s positive feedback, especially of how the theory predicts representational differences, inductive biases, and a BBP-like phase transition, and for their constructive and detailed comments, which we address below.
>
> ### **Equilibration of SGLD**
>
> It is correct that our theory assumes equilibration of SGLD. The minimization of the loss and the equilibration of the posterior happen in different time scales. However, the potentially slow time scale for equilibration does not appear to be critical for the qualitative phenomena we describe, since they persist in the absence of noise (see Fig. A, Fig. B in response to reviewer **c2BL**). Regarding the relation between SGD and SGLD, please also see our reply \[**SGLD**\] there.
>
> ### **Connection to state space models**
>
> We agree that extending the theory to gated RNNs and SSMs is an interesting direction. As SSMs such as S4 \[Gu & Dao, 2024\] encompass linear RNNs with shared weights, we hypothesized and tested that their inductive bias improves generalization in our regression setting (Fig. C in response to reviewer **c2BL**). An interesting future challenge will be to include gating into the present framework, possibly building on top of \[Krishnamurthy et. al, 2024\]. We have added this point to the discussion of the manuscript.
>
> ### **Generality of the theory**
>
> The theory is general: it applies to arbitrary activation functions and arbitrary datasets. Indeed, Figure 2 in the main text shows theory and experiments for a non-linear activation function $\\phi=erf$, giving exact predictions about the transition point ([Fig. D](https://anonymous.4open.science/r/rebuttal-16a25f/Landau_T4_erf.png)). As the reviewer points out, Figures 3-5 are implemented with a linear activation function. This choice was made because the theory becomes more interpretable. Similarly, regarding the tasks included in the current manuscript, they were chosen to clearly showcase the key mechanisms and predictions of the theory. We have created additional experiments for nonlinear activation and more complex tasks (see Figs. E, F, G in the reply to reviewer **pbAS**; see our reply **Empirical Validation** to reviewer **pbAS** and to reviewer **19Up**).
>
> ### **Relation to Clark et al. (2026)**
>
> At the time of submission, we were only aware of the results in the Cosyne abstract, Bordelon et al. (2024). During the review process these results appeared in a comprehensive pre-print, Clark et al. (2026). We agree that this necessitates thorough discussion in the manuscript, for instance as:
>
> “On a technical level, Clark et al. (2026) derives a closely-related feature learning theory based on the Bayesian posterior for RNNs in continuous time with a memory term, extending kernel-based approaches \[Seroussi ‘23, Fischer ‘24, Lauditi ‘25\] to the recurrent setting. In contrast, our theory works in discrete time with no memory term, since we aim to understand the functional implications of weight sharing by comparing DNNs and RNNs.
> Regarding generalization, their work derives a closed-form solution for the kernel for linear RNNs for stationary tasks in Fourier space. We use a perturbative analysis to describe how task information shapes the kernel.
> Regarding learning regimes, they find a transition in terms of an abstract feature-learning strength that suppresses chaos while we find a transition between temporally incoherent and temporally coherent representations controlled by the learning signal strength.“
> A priori, our transition is different, yet some signatures (such as spectral outliers) are shared. It would be very interesting to see how these phenomena are related.”
>
> ### **Computational complexity**
>
> The numerical solutions require kernel inversions that scale as $O((PT)^3)$ and incur significant cost due the number of MCMC samples in order to get stable gradient estimates. We added a section stating the complexity of the algorithm following its description in App. A.6 of the revised manuscript, following previous works such as Bordelon et al. (2022, Table 1), and Fischer et al. (2025, App. C.2).
>
> ### **Other clarifications**
>
> > Do the authors have any numerical solutions to the nonlinear versions of their equations? My impression was that *(it seems like the question got truncated?)*
>
> We describe the algorithm used for the numerics of the nonlinear theory in Section A.6: in short, we calculate gradients of the action $S(\\Phi, \\tilde C)$, and perform descent along $\\nabla\_\\Phi S$ and ascent along $\\nabla\_{\\tilde C} S$. Since $\\tilde C$ is a dual variable that is arg-*maximized*, we adopt an extragradient scheme to reduce oscillations around the joint saddle, and enforce positive definiteness of $\\Phi$ via a mirror-prox mapping of its gradient.
>
> > Are there theory curves on Fig. A.6?
>
> Thanks for pointing this out\! We have added the missing theory curves (Fig. D) and will include this updated version of the figure in the manuscript.

---

> > ### Author Rebuttal · Reviewer_G1f5 · 2026-04-01
> >
> > I appreciate the authors' detailed responses to my questions. I am in favor of acceptance and will increase my score!

---

### Official Review · Reviewer_c2LB · 2026-03-12

**Soundness:** 3
**Presentation:** 3
**Significance:** 2
**Originality:** 2
**Overall Recommendation:** 4
**Confidence:** 3

**Summary:**

This work develops a unified mean-field theory of feature learning in RNNs and DNNs in terms of representational kernels describing trained networks after convergence with Langevin dynamics. This theory casts network training as Bayesian inference over timesteps and patterns, revealing how architectural choices biases the functional form of the learned network. They use this theory to show (1) that for "endpoint-supervised" tasks like classification different phases of feature learning exist and (2) that for “sequential” tasks the weight sharing in RNNs leads to an inductive bias that enables more sample-efficient learning.

**Compliance With Llm Reviewing Policy:**

Affirmed.

**Final Justification:**

The rebuttal resolved most of my questions and reinforced my view that the paper is technically solid, clear, and likely useful to researchers interested in feature learning in recurrent architectures. I remain less convinced that the unified DNN/RNN framing is itself the main source of significance, and I still view the isotropic Langevin assumption as an important limitation, but these concerns do not outweigh the paper’s soundness and the novelty of its mean-field analysis of RNN feature learning, so my overall assessment remains positive.

**Key Questions For Authors:**

1. I found the definition of the DLN in Section 3.1 somewhat confusing. The architecture described there does not correspond to a standard feedforward DNN, since inputs appear to be injected into hidden layers. It would help if the introduction clarified earlier that the “unrolled” RNN is not equivalent to a standard feedforward DNN, but rather to a network that receives inputs at its hidden layers (as illustrated in Figure 1). This distinction only becomes clear later in the paper (bottom of page 4), but it would be helpful to explain it earlier.

2. Relatedly, does the unrolled architecture still assume weight sharing in the projection matrix mapping the input space to the hidden representation when inputs are injected at multiple layers?

3. What assumptions are made about the covariance structure of the gradient noise $K$ in the Langevin dynamics? Many works have shown that the noise in SGD is highly structured, anisotropic, and parameter dependent. Since the theory relies on Langevin dynamics as an approximation to SGD, it would be helpful to clarify what assumptions are made about $K$ and how sensitive the results are to this choice.

4. Beyond the phase transition toward temporal coherence and the inductive bias that improves generalization on sequential tasks, what additional insights does your theory provide about feature learning in RNNs? For instance, your discussion suggests connections to BBP-like transitions in the representational kernel and to the role of initialization scale in controlling feature learning. Could your theory generate further testable predictions—for example regarding the spectrum of learned representations, the role of pattern correlations, or the effect of nonlinearities—that could be validated empirically?

**Limitations:**

Yes.

**Strengths And Weaknesses:**

Strengths:

1. The paper carefully defines its notation and theoretical framework, and the results appear to be derived in a systematic and technically sound manner.

2. The mean-field analysis of feature learning in RNNs appears to be novel. While similar analyses have been performed for feedforward networks, extending this style of theoretical framework to recurrent architectures is an interesting direction.

3. The paper is generally well structured and walks the reader through the results in a logical order. The figures are clear and helpful in illustrating the main ideas.

Weaknesses

1. It is not entirely clear what new insight is gained by introducing a unified framework for DNNs and RNNs. The results do not appear to reveal fundamentally new properties of either architecture that were not already qualitatively understood. In particular, the conclusion that weight sharing in RNNs induces an inductive bias that enables more sample-efficient learning for sequential tasks is a fairly standard observation. The paper would be more compelling if it highlighted what new predictions or insights the theory provides beyond this qualitative understanding.

2. The most interesting aspect of the work appears to be the mean-field analysis of feature learning in RNNs. Emphasizing the implications of this analysis more directly—rather than focusing primarily on the unifying framework—might increase the significance of the paper.

---

> ### Author Rebuttal · Authors · 2026-03-31
>
> We thank the reviewer for their positive assessment of the relevance of the mean-field theory for RNNs, and of the structure and soundness of the presentation. Below we address the specific comments:
>
> ### **Relevance of *unified* theory**
>
> The broad goal of our work is to connect architectural structure to functional inductive biases. An important structural feature is weight sharing. While we agree that a mean-field theory of RNNs is interesting and new, we believe that a unified framework is essential to highlight the consequences of weight sharing by direct comparison of architectures.
>
> ### **Empirically verifiable predictions from theory**
>
> The primary benefit of our theory is that it abstracts away high-dimensional weights in favor of computational parameters of the network: the neural representation $h$ and predictor $f$, facilitated by the kernel (Eq. (3)). This approach allows us to relate two areas: gradient-descent based training of networks (e.g., RNNs), and probabilistic models of representation (the Bayesian posterior, specifically Gaussian processes such as the LGSSM), which enables specific predictions:
>
> - **Learning phases**: Only a strong learning signal will induce a temporally coherent representation in RNNs, identifying two distinct regimes. Beyond theoretical interest, this practically suggests choosing small initialization *in addition to* $\\mu$P scaling in RNNs for improved generalization: a scaling rule that, to our knowledge, has not been identified before.More broadly, it opens up a search for similar transitions in other architectures, potentially complementing recent findings \[Tu \\& Aranguri \\& Jacot 2024\] in deep networks.
> - **Representation spectrum**: Our theory predicts a new signature of strong feature learning (temporal coherence) that could be probed in artificial or biological networks: it is not sufficient that the hidden representation is just rescaled under learning. Rather, strong learning necessarily produces spectral outliers in the kernel, with corrections of the form $\\Phi \+ \\epsilon yy^T$. Each kernel eigenvalue in $\\Phi v\_\\mu \= \\lambda\_\\mu v\_\\mu$ changes by $\\epsilon v\_\\mu^T yy^T v\_\\mu$. For isotropic $\\Phi$, this yields an $O(P)$ change in the target direction $y$, since $\\epsilon yy^T y \= \\epsilon P y$. Thus, the adapted kernel selectively reduces discrepancies between target $y$ and network output $f$.
> - **Generalization/pattern correlations**: The kernel is the sufficient statistic to obtain the network’s predictions after learning. By showing that weight sharing induces temporal correlations in the kernel  $\\Phi$, the theory predicts that this property is useful specifically for learning temporally correlated sequences, an insight that can carry over to other architectures. We have tested this hypothesis for state-space models and transformers ([Fig. C](https://anonymous.4open.science/r/rebuttal-16a25f/RNNs_are_efficient_sequence_learners_all_arch.png)), which employ similar weight sharing. See also our task-model alignment reply to **pbAS.**
>
> > **\[SGLD\]** What assumptions are made about the covariance structure of the gradient noise K in the Langevin dynamics?
>
> Strictly speaking, our theory and SGLD describe the equilibrated Bayesian posterior over parameters with isotropic noise $\\mathbf{K}\\propto \\mathbf{I}$ (App. A3 & A4), which can be seen as a naive approximation of SGD \[Welling \\& Teh, 2011\]. Yet the qualitative phenomena we identify, phase transition and generalization, persist even when noise is set to zero (see [Fig. A](https://anonymous.4open.science/r/rebuttal-16a25f/RNNs_vs_DNNs_endpoint_sv__kappa_0.png), [Fig. B](https://anonymous.4open.science/r/rebuttal-16a25f/RNNs_are_efficient_sequence_learners__kappa_0.png)), suggesting that at least these results extend to plain gradient descent. Still, there certainly are many phenomena in neural network training that do depend on the structure of gradient noise, and we agree that incorporating this into the framework is an exciting direction for future work.
>
> #### *Other comments*
>
> > I found the definition of the DLN in Section 3.1 somewhat confusing.
>
> We introduced this 'generalized' DNN, which receives inputs and produces outputs at each layer, to enable direct comparison with the RNN's input-output setting and isolate the effect of weight sharing. The revision now explains this choice. For Fig. 2, we reduce it to the standard endpoint-supervised version of DNNs. Input $\\mathbf{U}$ and readout $V$ weights are always shared across time (Fig. 1\) to keep the comparison tight, but this could be easily relaxed.

---

> > ### Author Rebuttal · Reviewer_c2LB · 2026-04-03
> >
> > Thank you for your rebuttal. I remain unconvinced that a unified theory is necessary for making the paper’s predictions useful or insightful, and I would still emphasize the isotropic assumption in the Langevin dynamics as a weakness that should be stated more clearly in the main text.
> >
> > That said, most of my concerns and questions have been resolved, and I remain positive about the work overall. I will keep my original score.

---

> > > ### Author Response · Authors · 2026-04-04
> > >
> > > ### Unified theory
> > >
> > > We agree that a description of feature learning in RNNs does not require a unified framework. However, our goal here is describe the representational and functional implications of architecture, in particular weight sharing. Without a shared framework, the results for each architecture lack a natural baseline, and would be hard to interpret. For instance, we think Fig. 5 left + Eq. 9 [DNN] / Fig. 5 right + Eq. 10 [RNN] in the manuscript would be hard to interpret if only either architecture was presented. Relatedly, we believe this facilitates the extension to SSMs and transformers, as in the new [Fig. C](https://anonymous.4open.science/r/rebuttal-16a25f/RNNs_are_efficient_sequence_learners_all_arch.png), be it for future theoretical work that addresses architecture specifics such as neural scaling laws. On a more practical side, a unified theory can help to choose one architecture over the other in practice depending on the task at hand.
> > >
> > >
> > > ### SGLD
> > >
> > > To clarify our initial reply: we do not claim that our theory faithfully describes anisotropic SGD noise, but the *specific* qualitative implications in the manuscript do not depend on it, as they persist in absence of any noise $\kappa \rightarrow 0$ (added [Fig. A](https://anonymous.4open.science/r/rebuttal-16a25f/RNNs_vs_DNNs_endpoint_sv__kappa_0.png), [Fig. B](https://anonymous.4open.science/r/rebuttal-16a25f/RNNs_are_efficient_sequence_learners__kappa_0.png)).
> > >
> > > We would also like to point out that the limit $\kappa \rightarrow 0$ and jointly the weight decay term $\frac{\kappa}{G_\theta}\rightarrow 0$ can be performed in a well-behaved manner, such that the prior variance stays constant; thus the theory contains the important case of full batch gradient flow without noise.
> > >
> > > Without doubt there are phenomena where the anisotropy of SGD noise is crucial, like [escape time from local minima](https://proceedings.mlr.press/v97/zhu19e.html). We would be eager to learn and test what other phenomena seem likely to require an extended theory.
> > >
> > > Especially since reviewers **G1f5** and **19Up** have expressed similar comments, we have now detailed this limitation in the manuscript and would be happy if you have feedback:
> > >
> > > In **Section 3.1**:
> > > > Note that in contrast to SGD where stochasticity arises from fluctuations between mini-batches, SGLD assumes noise that is i.i.d. across components and updates.
> > >
> > > In **Limitations**:
> > > > Regarding training, we approximated the correlated mini-batch fluctuations in plain SGD with isotropic noise of SGLD (Welling and Teh, 2011). Whereas the qualitative results discussed here persist in the noiseless case (see Appendix A.8 <containing the rebuttal Figs. A,B>), this replacement in general changes optimization trajectories. Note that the limit $\kappa\rightarrow 0$  and jointly the weight decay term $\frac{\kappa}{G_\theta}\rightarrow 0$ can be taken such that the prior variance stays constant and the theory is still well behaved, so that it also allows the study of full batch gradient flow.
> > >
> > > ---
> > > Please let us know whether this addresses your questions, and if concerns remain.

---

### Decision · Program_Chairs · 2026-04-30

**Decision:**

Accept (regular)

**Comment:**

The authors studied feature learning regime via the lens of Bayesian neural networks for RNNs and DNNs. The main contribution is a unified theory for mean-field/kernel in the $\mu$P regime that poses the effect of weight sharing as a masking operation. A phase transition toward temporal coherence and improve generalization is interesting as well.

Reviewers generally found the theory for RNN the most significant contribution, albeit the unification is less interesting. Some concerns were raised regarding presentation, both clarity as well as positioning of related works. Reviewers were also skeptical about the approach of studying the stationary distribution of SGLD as a proxy for end of training, which are valid concerns. The authors addressed most of these concerns, albeit some are more fundamental to the approach, like the Bayesian framework.

Overall, I find all reviewers eventually found this submission a positive contribution, with none of the remaining concerns sufficiently critical to the evaluation of this work. Therefore, conditional on the authors incorporating the rebuttal changes and clarifications into the final version, I would recommend accept for this paper.